# A Memory-Efficient Hierarchical Algorithm for Large-scale Optimal Transport Problems

**Wenzhou Xia**[1], **Ya-Nan Zhu**[2], **Jingwei Liang**[1,3,†], **Xiaoqun Zhang**[1,3,4,†]

## Abstract

We propose `HALO`, a memory-efficient hierarchical algorithm for solving large-scale optimal transport (OT) problems with squared Euclidean cost, particularly effective in moderate-dimensional settings. The core of `HALO` lies in combining a hierarchical representation of the OT problem with parallel-friendly linear programming solvers, within which an active pruning technique is integrated to further reduce memory usage and computational cost. Theoretically, we establish a scale-independent iteration-complexity upper bound for the refinement phase, which is consistent with our numerical observations. Numerically, experiments on the image dataset DOTmark and the 3D point cloud dataset ModelNet10 demonstrate that `HALO` effectively alleviates the memory and scalability bottlenecks of existing solvers. Our method demonstrates significant advantages compared to state-of-the-art baselines: for images with $n = 1024^2$ pixels, it achieves an $8.9\times$ speedup and $70.5\%$ reduction in memory usage under comparable accuracy; for 3D point clouds at scale $n = 2^{18}$, it achieves a $1.84\times$ speedup and an $83.2\%$ reduction in memory usage with $24.9\%$ lower transport cost.

## 1 Introduction

The Wasserstein distance, defined via the Kantorovich formulation of Optimal Transport (OT) problems, is a powerful metric for measuring the similarity between two probability distributions. It has been widely adopted in various fields such as generative modeling Arjovsky et al. (2017); Kornilov et al. (2024); Tong et al. (2023); Hui et al. (2025), color transfer Solomon et al. (2015); Pitié & Kokaram (2007), texture synthesis and mixing Dominitz & Tannenbaum (2009); Rabin et al. (2011), registration and deformation Haker et al. (2004); Rehman et al. (2007), image restoration He et al. (2021), domain adaptation Montesuma & Mboula (2021); He et al. (2024) and transportation-based morphology metrics Basu et al. (2014). However, modern large-scale applications push OT solvers to regimes where classical formulations become computationally prohibitive and memory intensive.

Given probability measures $\mu, \nu$ on domains $\mathbb{S}$ and $\mathbb{D}$ with ground cost $c : \mathbb{S} \times \mathbb{D} \to \mathbb{R}$, the Kantorovich problem is defined by

$$\inf_{\pi \in \Pi(\mu,\nu)} \int_{\mathbb{S} \times \mathbb{D}} c(s, d) \, \mathrm{d}\pi(s, d), \tag{1}$$

where $\Pi(\mu, \nu)$ denotes the set of couplings on $\mathbb{S} \times \mathbb{D}$ with marginals $\mu$ and $\nu$ on $\mathbb{S}$ and $\mathbb{D}$, respectively.

In the discrete case, the supports are $\mathbb{S} = \{s_1, \ldots, s_m\}$ and $\mathbb{D} = \{d_1, \ldots, d_n\}$, with corresponding probability vectors $\boldsymbol{u} \in \mathbb{R}^m$ and $\boldsymbol{v} \in \mathbb{R}^n$ defined by $u_i = \mu(s_i)$ and $v_j = \nu(d_j)$. A coupling is represented by a nonnegative matrix $\boldsymbol{X} \in \mathbb{R}_+^{m \times n}$ whose row and column sums equal $\boldsymbol{u}$ and $\boldsymbol{v}$. The ground cost is encoded in the cost matrix $\boldsymbol{C} = (c(s_i, d_j)) \in \mathbb{R}^{m \times n}$. Consequently, this setup leads to the following discrete OT problem in standard Linear Programming (LP) form:

$$\min_{\boldsymbol{x} \geq 0} \boldsymbol{c}^\top \boldsymbol{x}, \quad \text{s.t.} \quad \boldsymbol{A}\boldsymbol{x} = \boldsymbol{q}, \tag{2}$$

[1] School of Mathematical Sciences, Shanghai Jiao Tong University, Shanghai, China; [2] School of Mathematics, Harbin Institute of Technology, Harbin, China; [3] Institute of Natural Sciences, Shanghai Jiao Tong University, Shanghai, China; [4] Shanghai-Chongqing Institute of Artificial Intelligence, Shanghai, China. Emails: `xwz.hhh@sjtu.edu.cn`, `yananzhu6@gmail.com`, `jingwei.liang@sjtu.edu.cn`, `xqzhang@sjtu.edu.cn`. [†] Corresponding authors.

where $\boldsymbol{x} = \text{vec}(\boldsymbol{X})$, $\boldsymbol{c} = \text{vec}(\boldsymbol{C})$, $\boldsymbol{q}^\top = (\boldsymbol{u}^\top, \boldsymbol{v}^\top)$, and $\boldsymbol{A} = \begin{bmatrix} \mathbf{1}_n^\top \otimes \boldsymbol{I}_m \\ \boldsymbol{I}_n \otimes \mathbf{1}_m^\top \end{bmatrix} \in \mathbb{R}^{(m+n) \times mn}$.

This LP problem has $mn$ variables and $m + n$ constraints. Considering OT problems between two images of size $n = r \times r$, the variable count in $\boldsymbol{X}$ scales as $n^2$, reaching approximately $10^{12}$ when $r = 1024$. Such an enormous scale results in prohibitive computational cost and memory consumption. Moreover, traditional solvers such as the network simplex Gabow & Tarjan (1991) and interior-point Pele & Werman (2009) methods further fail to exploit the modern GPU architectures, leading to a growing gap between classical OT algorithms and the scalability demanded in contemporary applications.

**Related work** Existing algorithms for large-scale OT fall into three broad categories:

The first category consists of approximation methods, which trade accuracy for efficiency. Representative examples include entropy-regularized algorithms such as Sinkhorn Cuturi (2013); Dvurechensky et al. (2018); Schmitzer (2019); Chen et al. (2022), low-rank algorithms Scetbon et al. (2021); Halmos et al. (2025) as well as approaches based on approximate metrics such as the approximated earth mover's distance Shirdhonkar & Jacobs (2008) and the sliced Wasserstein distance Kolouri et al. (2019); Nadjahi et al. (2021). These methods achieve good scalability but at the cost of reduced accuracy.

The second category comprises LP-based algorithms, which aim to accelerate OT by improving LP solvers tailored to this structured problem. Representative methods include randomized block coordinate descent methods Xie et al. (2024), semi-smooth Newton-type approaches Li et al. (2020), and first-order algorithms such as Douglas–Rachford splitting Mai et al. (2021), dual extrapolation Jambulapati et al. (2019), Halpern–Peaceman–Rachford methods Zhang et al. (2022); Chen et al. (2024), Halpern iteration Zhang et al. (2025), and primal–dual hybrid gradient (PDHG) Esser et al. (2010); Chambolle & Pock (2011). Among them, factorization-free first-order LP solvers, such as PDHG and its variants Applegate et al. (2021); Lu et al. (2023; 2025), are particularly attractive for large-scale OT, since their computations are dominated by matrix-vector multiplications and are therefore well suited for GPU parallelism. Nevertheless, despite these advances, LP-based solvers continue to suffer from severe memory bottlenecks in ultra-large-scale settings Lu & Yang (2024).

The third category consists of multiscale algorithms that exploit the pyramid structure of OT Mérigot (2011); Gerber & Maggioni (2017); Leclaire & Rabin (2019), with representative methods such as `ShortCut` Schmitzer (2016) and the multiscale semi-smooth Newton (MSSN) method Liu et al. (2022). Although these methods effectively reduce computational cost in practice, the lack of iteration-complexity bounds leaves their worst-case efficiency unresolved, and the dependency on CPU-based LP solvers limits their scalability on modern parallel hardware.

Overall, developing OT solvers that simultaneously meet the demands of memory efficiency, large-scale parallel computation, and high accuracy remains an open challenge.

**Main contributions** We propose `HALO`, a **memory-efficient** and **parallel-friendly** **H**ierarchical **A**lgorithm for **L**arge-scale **O**ptimal **T**ransport problems with squared Euclidean cost, which is particularly effective in moderate-dimensional settings. The key idea is to exploit the hierarchical structure of OT by solving a hierarchy of OT problems across resolutions: coarse-level solutions warm-start finer levels, and each level alternates between updating active support and solving a restricted LP on the active support with GPU-friendly solvers.

The significance of `HALO` lies in the following aspects:

- *Memory efficiency:* `HALO` requires only $\mathcal{O}(n)$ memory, matching (to the best of our knowledge) the lowest space complexity among existing GPU-based OT solvers.
- *Parallelism:* `HALO` uses LP solvers based on factorization-free first-order methods, whose dominant operations are matrix-vector multiplications, enabling efficient parallel implementation.
- *Theory:* Under certain assumptions, `HALO` enjoys a scale-independent iteration-complexity bound per level, which is also validated empirically.
- *Empirical performance:* On DOTmark at $n = 1024^2$, `HALO` achieves an $8.9\times$ speedup and a $70.5\%$ GPU-memory reduction; on ModelNet10 with $n = 2^{18}$, it achieves a $1.84\times$ speedup, an $83.2\%$ GPU-memory reduction, and a $24.9\%$ lower transport cost.

**Organization**   The rest of the paper is organized as follows: In Section 2, we introduce the proposed algorithm, where Section 2.3 provides the scale-independent iteration-complexity bound. In Section 3 we present the experimental results, and Section 4 concludes the paper.

**Notation**   $\mathbb{S}, \mathbb{D}$ denote the source and target spaces in OT problem. In discrete setting, given $\mathbb{N} = \{(s_{i_k}, d_{j_k}) \mid 1 \le k \le K\} \subset \mathbb{S} \times \mathbb{D}$, $\boldsymbol{A}_{\mathbb{N}}$ denotes the sub-matrix of the constraint matrix $\boldsymbol{A}$ restricted to the columns indexed by $\mathbb{N}$, and $\boldsymbol{x}_{\mathbb{N}}$ as the corresponding sub-vector of the vector $\boldsymbol{x} \in \mathbb{R}^{nm}$. Let $\mathrm{Top}_K(\mathbb{C})$ denote the operator that selects $K$ pairs from the set $\mathbb{C} \subset \mathbb{S} \times \mathbb{D}$, where these pairs have the top $K$ associated values.

## 2   HALO: A HIERARCHICAL ALGORITHM FOR LARGE-SCALE OT

### 2.1   OUTLINE OF THE PROPOSED FRAMEWORK

We first outline the overall workflow of HALO. The key reason behind the efficiency of HALO lies in two aspects: (i) an inherent multiscale structure that enables a coarse-to-fine hierarchical representation, and (ii) sparsity of the transport plan which allows us to incorporate active-support detection to the solver. These two aspects jointly contribute to the significant reduction of the computational complexity and memory demand.

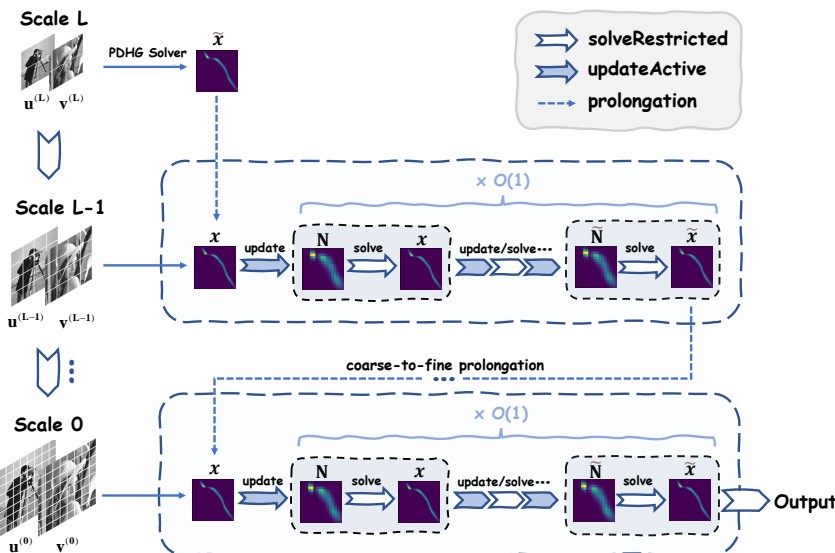

Figure 1: Architecture of HALO.

Figure 1 provides a high-level overview of our HALO method:

• Hierarchical Structure (Outer Loop over Levels): We construct an $(L + 1)$-level hierarchy of OT problems, ranging from the coarsest $L$-th level to the finest 0-th level. For each level $\ell \in \{0, 1, ..., L\}$, let $(\boldsymbol{u}^{(\ell)}, \boldsymbol{v}^{(\ell)}, \boldsymbol{c}^{(\ell)})$ denote the marginals and costs, and $(\boldsymbol{f}^{(\ell)}, \boldsymbol{g}^{(\ell)})$ denote the dual potentials. Based on the LP formulation (2), the equivalent min–max formulation of the OT problem at the $\ell$-th level reads:

$$\min_{\boldsymbol{x}^{(\ell)} \in \mathbb{R}_+^{m_\ell n_\ell}} \max_{\boldsymbol{f}^{(\ell)} \in \mathbb{R}^{m_\ell}, \boldsymbol{g}^{(\ell)} \in \mathbb{R}^{n_\ell}} \langle \boldsymbol{u}^{(\ell)}, \boldsymbol{f}^{(\ell)} \rangle + \langle \boldsymbol{v}^{(\ell)}, \boldsymbol{g}^{(\ell)} \rangle + \langle \boldsymbol{c}^{(\ell)}, \boldsymbol{x}^{(\ell)} \rangle - \langle \boldsymbol{A}^{(\ell)} \boldsymbol{x}^{(\ell)}, (\boldsymbol{f}^{(\ell)}, \boldsymbol{g}^{(\ell)}) \rangle.$$

$$(3)$$

where $\boldsymbol{A}^{(\ell)}$ is the OT constraint matrix at level $\ell$, and $m_\ell, n_\ell$ denote the marginal scales. We leverage this hierarchy via a coarse-to-fine strategy: For $\ell = L$, solve problem (3) and denote the solution as $(\boldsymbol{x}^{(L)}, \boldsymbol{f}^{(L)}, \boldsymbol{g}^{(L)})$; for $\ell \in \{0, \cdots, L - 1\}$, the solution $(\boldsymbol{x}^{(\ell+1)}, \boldsymbol{f}^{(\ell+1)}, \boldsymbol{g}^{(\ell+1)})$ from the coarser $(\ell + 1)$-th level is prolongated to initialize $(\boldsymbol{x}^{(\ell)}, \boldsymbol{f}^{(\ell)}, \boldsymbol{g}^{(\ell)})$ for the finer $\ell$-th level; see Section 2.2 for details.

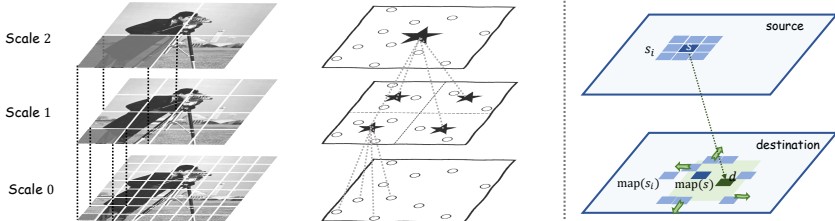

Figure 2: Left & Middle: Hierarchy construction of images and point clouds. Right: Shielding in `updateActive`; green region indicates possible unshielded $d$, and arrows denote pairs $(s, d)$ added by the shielding-based component.

- Refinement Iteration (Inner Loop at a Fixed Level): At each specific level $\ell \in \{0, \cdots, L-1\}$, we refine the initialized solution to solve the OT problem (3). To fully exploit the sparsity of the transport plan, we alternate between updating the active support and solving the corresponding LP problem using GPU-based solvers (instantiated by PDHG-based algorithm Lu et al. (2025)); see Section 2.3 and Algorithm 2 for details.

Summarizing the above discussions leads to the proposed `HALO` method.

---

**Algorithm 1** `HALO`: **H**ierarchical **A**lgorithm for **L**arge-scale **O**ptimal Transport

---

1: **Input**: marginal distributions $\boldsymbol{u}$ and $\boldsymbol{v}$, ground cost $\boldsymbol{c}$, number of levels $L$.
2: **Output**: optimal coupling $\boldsymbol{x}^{(0)}$.
3: Build hierarchical OT problems $\{(\boldsymbol{u}^{(\ell)}, \boldsymbol{v}^{(\ell)}, \boldsymbol{c}^{(\ell)})\}_{\ell=0}^{L}$.
4: Solve OT with $(\boldsymbol{u}^{(L)}, \boldsymbol{v}^{(L)}, \boldsymbol{c}^{(L)})$ on the coarsest level $L$ to obtain $(\boldsymbol{x}^{(L)}, \boldsymbol{f}^{(L)}, \boldsymbol{g}^{(L)})$.
5: **for** $\ell = L-1, \ldots, 0$ **do**
6:     Initialize $(\boldsymbol{x}^{(\ell)}, \boldsymbol{f}^{(\ell)}, \boldsymbol{g}^{(\ell)})$ by prolongating $(\boldsymbol{x}^{(\ell+1)}, \boldsymbol{f}^{(\ell+1)}, \boldsymbol{g}^{(\ell+1)})$.
7:     **repeat**
8:       **1). Updating active support**, detailed in Algorithm 3
9:           $\mathbb{N}^{(\ell)} \leftarrow \texttt{updateActive}(\boldsymbol{x}^{(\ell)}, \boldsymbol{f}^{(\ell)}, \boldsymbol{g}^{(\ell)}, \mathbb{N}^{(\ell)})$
10:       **2). Solving restricted OT**, detailed in Section 2.3
11:           $(\boldsymbol{x}^{(\ell)}, \boldsymbol{f}^{(\ell)}, \boldsymbol{g}^{(\ell)}) \leftarrow \texttt{solveRestricted}(\boldsymbol{x}^{(\ell)}, \boldsymbol{f}^{(\ell)}, \boldsymbol{g}^{(\ell)}; \mathbb{N}^{(\ell)}, \boldsymbol{u}^{(\ell)}, \boldsymbol{v}^{(\ell)}, \boldsymbol{c}^{(\ell)})$
12:     **until** $\boldsymbol{x}^{(\ell)}$ meets the termination criteria.
13: **end for**
14: **return** $\boldsymbol{x}^{(0)}$.

---

For the rest of this section, we discuss the details of the key steps in Algorithm 1, including the construction of the hierarchy, the active-support updating strategy and its convergence property.

## 2.2 HIERARCHY: COARSE-TO-FINE MULTISCALE STRUCTURE

A high-quality initialization greatly accelerates the inner loop in Algorithm 1. To this end, we exploit the geometry of OT and compute such an initialization via a coarse-to-fine multiscale scheme.

Consider a discrete OT problem with supports $\mathbb{S} = \{s_1, \cdots, s_m\}, \mathbb{D} = \{d_1, \cdots, d_n\}$ and marginals $\mu = \sum_{i=1}^{m} u_i \delta_{s_i}, \nu = \sum_{j=1}^{n} v_j \delta_{d_j}$. In our hierarchy, the supports at the $\ell$-th level are formed by representative points of corresponding neighbor groups from the $(\ell-1)$-th level. Accordingly, the marginals $\boldsymbol{u}^{(\ell)}$ and $\boldsymbol{v}^{(\ell)}$ are obtained by aggregating the masses of these constituent groups, and the cost $\boldsymbol{c}^{(\ell)}$ is defined by the distance between representative points.

As illustrated in Figure 2, in the image setting where $\mathbb{S}$ and $\mathbb{D}$ correspond to pixels on a regular grid, the hierarchy is built by recursively merging $2 \times 2$ pixels, using the barycenter of each block as its representative. In non-grid settings (e.g., point clouds), we instead employ spatial partitioning structures, such as $2^d$-trees or kd-trees Finkel & Bentley (1974); Bentley (1975), to construct the hierarchy, as detailed in Appendix C.1.

## 2.3 Sparsity: Solve Restricted OT on Active Support

Classical LP theory Bertsimas & Tsitsiklis (1997) states that the optimal solution $x^* \in \mathbb{R}^{mn}$ of problem (2) has at most $m + n$ nonzero entries, which is sparse when $m, n$ are large. If the support $\text{supp}(x^*)$ were known a priori, we could simply solve the problem restricted to this set. Therefore, letting active support $\mathbb{N} \subset \mathbb{S} \times \mathbb{D}$ be an estimation of $\text{supp}(x^*)$, we define the restricted OT problem on $\mathbb{N}$ as follows:

**Definition 1 (Restricted OT on active support $\mathbb{N}$)**

$$\min_{x} \; c_{\mathbb{N}}^\top x_{\mathbb{N}} \quad s.t. \quad A_{\mathbb{N}} x_{\mathbb{N}} = q, \;\; x_{\mathbb{N}} \geq 0 \;\; x_{\mathbb{N}^{\complement}} = 0, \tag{4}$$

where $A_{\mathbb{N}}$ is the submatrix of $A$ restricted to columns indexed by $\mathbb{N}$, $c_{\mathbb{N}}$ is the corresponding subvector of $c$, $x_{\mathbb{N}}$ and $x_{\mathbb{N}^{\complement}}$ are the corresponding subvectors on $\mathbb{N}$ and $\mathbb{N}^{\complement}$.

**Remark 1** *Since the OT problems at each level share the same formulation and differ only in scale, we omit the level superscript $(\ell)$ and simply use the form (2) for illustration.*

If the active support $\mathbb{N}$ exactly corresponds to $\text{supp}(x^*)$, solving the restricted problem provides the global optimizer of the original OT problem. However, finding $\text{supp}(x^*)$ is as challenging as solving the problem. To address this difficulty, we consider Algorithm 2, a refinement procedure that alternates between updating the active support based on the current solution and solving the restricted OT on the newly updated active support.

---

**Algorithm 2** A refinement at each level

INPUT: initial coupling $x^0$ and dual potentials $(f^0, g^0)$; $u, v$ and $c$.
OUTPUT: global optimizer $x$

$k \leftarrow 0$, $\mathbb{N}^0 \leftarrow \emptyset$
**repeat**
   $\mathbb{N}^{k+1} \leftarrow \texttt{updateActive}(x^k, f^k, g^k, \mathbb{N}^k)$
   $(x^{k+1}, f^{k+1}, g^{k+1}) \leftarrow \texttt{solveRestricted}(x^k, f^k, g^k, \mathbb{N}^{k+1}; u, v, c)$
   $k \leftarrow k + 1$
**until** $x^k$ meets the termination criteria
**return** $(x^k, \mathbb{N}^k)$

---

The efficiency of Algorithm 2 hinges on the design of `updateActive`. In previous work, `ShortCut` Schmitzer (2016) constructs a new active support by augmenting the support of the previous coupling with a **shielding-based component**. This component relies on the shielding condition, defined as follows:

**Definition 2** *(Shielding condition) Given $\mathbb{S}$, $\mathbb{D}$ and cost $c : \mathbb{S} \times \mathbb{D} \to \mathbb{R}$. Let $s, s' \in \mathbb{S}, d, d' \in \mathbb{D}$. We say $(s', d')$ shields $s$ from $d$ if*

$$c(s, d) + c(s', d') > c(s, d') + c(s', d).$$

To obtain a provable $\mathcal{O}(1)$ per-level iteration bound and improve robustness in practice, we introduce two modifications: (i) expanding the active support to include the entire previous ones, rather than only the previous coupling's support, and (ii) performing dual-violation correction, adding pairs suggested by significant violations in the dual potentials. The complete update procedure is summarized in Algorithm. 3.

Although the first improvement makes the estimated active support larger, it provides a scale-independent iteration-complexity bound, as detailed in Theorem 1, ensuring that the growth of the active support does not affect its sparsity. Additionally, in the implementation, we use the factorization-free and parallel-friendly LP solvers, which is less sensitive to the problem scale compared to traditional LP solvers. As a result, our algorithm achieves excellent practical performance while benefiting from a theoretical iteration-complexity guarantee.

The second improvement involves a dual-violation correction mechanism that accelerates convergence by adding pairs suggested by significant violations in the dual potentials. This approach

ensures that the active support includes $\text{supp}(\boldsymbol{x}^*)$ as much as possible, while maintaining sparsity. As a result, the algorithm refines the solution more efficiently, particularly in some challenging instances. Unlike MSSN Liu et al. (2022) which relies on a threshold parameter, we employ a $\text{Top}_K$ operator to select pairs with the largest dual violations, ensuring the active support remains sparse throughout.

---

**Algorithm 3** `updateActive`

---

INPUT: current coupling $\boldsymbol{x} = \text{vec}(\boldsymbol{X})$, dual potentials $(\boldsymbol{f}, \boldsymbol{g})$, cost $\boldsymbol{c} \in \mathbb{R}^{nm}$ with $c : \mathbb{S} \times \mathbb{D} \to \mathbb{R}$, active support $\mathbb{N}$, dual-violation hyperparameter $\beta$
OUTPUT: updated active support $\mathbb{N}'$

**1). start from current active support**
$\mathbb{N}' \leftarrow \mathbb{N}$

**2). shielding-based**
construct $\text{map} : \mathbb{S} \to \mathbb{D}$, $\text{map}(s_i) = \arg\max_{d_j \in \mathbb{D}} \boldsymbol{X}_{ij}$ for all $s_i \in \mathbb{S}$
Define $\mathcal{R}(s) \subset \mathbb{S}$ as the local neighborhood of $s$ (8-neighbors for images and KNN for point clouds, see Appendix C.2).
**for** $s \in \mathbb{S}$ **do**
    $D(s) \leftarrow \{\, \text{map}(s') : s' \in \mathcal{R}(s) \,\}$
    $\widehat{D}(s) \leftarrow \{\, d \in \mathbb{D} : d \text{ is not shielded from } s \text{ by } (s', \text{map}(s')), \ \forall s' \in \mathcal{R}(s) \,\}$
    **if** $\widehat{D}(s) = \varnothing$ **then**
        $\widehat{D}(s) \leftarrow \arg\min_{d \in \mathbb{D} \setminus D(s)} c(\text{map}(s), d)$
    **end if**
    $D(s) \leftarrow D(s) \cup \widehat{D}(s)$
    $\mathbb{N}' \leftarrow \mathbb{N}' \cup \{(s, d) : d \in D(s)\}$
**end for**

**3). dual–violation correction**
Pricing problem, $\mathbb{C} \leftarrow \{(s_i, d_j) \in \mathbb{S} \times \mathbb{D} : f_i + g_j > c_{ij}\}$
$K \leftarrow \beta \, |\mathbb{S}|, \quad \mathbb{C}^K \leftarrow \text{Top}_K(\mathbb{C})$
$\mathbb{N}' \leftarrow \mathbb{N}' \cup \mathbb{C}^K$

**return** $\mathbb{N}'$

---

**Theorem 1 (Scale-independent iteration-complexity bound)** *Let $(\mathbb{S}, \mathbb{D}, \mu, \nu)$ be a discrete OT instance with squared Euclidean ground cost. Consider Algorithm 2 with the update rule from Algorithm 3, which produces solutions $\{(\boldsymbol{x}^k, \boldsymbol{f}^k, \boldsymbol{g}^k)\}_{k \geq 0}$ and active supports $\{\mathbb{N}^k\}_{k \geq 0}$ via the monotone update*

$$\mathbb{N}^{k+1} \leftarrow \texttt{updateActive}(\boldsymbol{x}^k, \mathbb{N}^k).$$

*Assume there exist constants $q \in (0, 1]$, $D < \infty$, $\rho < \infty$, $L < \infty$, $R_0 < \infty$ such that:*

1. *(Directional coverage) For $\forall s \in \mathbb{S}$ and $\forall v \in \mathbb{R}^n$, $\exists s' \in S(s) \subset \mathbb{S}$ with $\langle v, \ s' - s \rangle \geq \|v\| \|s' - s\| q$.*

2. *(Bounded radius of $S(s)$) $\|s' - s\| < D$ for all $s$ and all $s' \in S(s)$.*

3. *(Bounded density of $\mathbb{D}$) $|\mathbb{D} \cap B_R(z)| \leq \rho \, \text{vol}_n(B_R)$ for all $z \in \mathbb{R}^n$ and $R > 0$.*

4. *(Uniform Lipschitz regularity) $\|\text{map}_k(s) - \text{map}_k(s')\| \leq L\|s - s'\|, \forall s, s' \in \mathbb{S}, k \geq 0$.*

5. *(Coupling stability) $\|\text{map}_k(s) - \text{map}_0(s)\| \leq R_0$ for all $k$ and all $s \in \mathbb{S}$.*

*Then there exists a constant $C > 0$ such that $\boldsymbol{x}_k$ is a global optimizer for all $k \geq C$, and $|\mathbb{N}_k| \leq C \, |\mathbb{S}|$ for all $k \geq 0$.*

**Remark 2** *Assumptions 1–3 hold in our setting; by contrast, Assumptions 4–5 are stronger yet natural under the multiscale warm-start. Since each level refines a warm start prolongation from the previous coarser level, the per-level updates are mild, making a uniform Lipschitz condition and a bounded-drift stability assumption intuitively justified. The proof of Theorem 1 is deferred to Appendix B.1.*

At the end of this subsection, we note that in Algorithm 2 the restricted OT on the active support $\mathbb{N}$ is solved by `solveRestricted`, which dominates computational cost and memory usage, making the LP solver choice critical. To achieve both memory efficiency and parallelism, we adopt a factorization-free first-order method, and we highlight the following property of the restricted OT under Pock–Chambolle rescaling Pock & Chambolle (2011):

**Proposition 1** *Let $\boldsymbol{B}$ be the constraint matrix of restricted OT problem, and define the rescaled matrix $\widetilde{\boldsymbol{B}}$ by the Pock–Chambolle rescaling:*

$$D_r = \operatorname{diag}\big(\sqrt{r_1}, \ldots, \sqrt{r_m}, \sqrt{c_1}, \ldots, \sqrt{c_n}\big), \quad D_c = \sqrt{2}\,\boldsymbol{I}, \quad \widetilde{\boldsymbol{B}} = D_r^{-1}\,\boldsymbol{B}\,D_c^{-1},$$

*with $r_i, c_j$ the row/column degrees of $\boldsymbol{B}$. Then $\|\widetilde{\boldsymbol{B}}\|_2 = 1$.*

This result justifies the use of a constant stepsize in PDHG-based algorithm, eliminating the need for norm estimation and reducing overhead in `HALO`. The proof is provided in Appendix B.2.

## 3 EXPERIMENTS

To evaluate the scalability and efficiency of `HALO`, we conduct experiments on two representative datasets: the 2D image dataset DOTmark Schrieber et al. (2016) and the 3D point cloud dataset ModelNet10 Wu et al. (2015). For image experiments, we compare `HALO` against three state-of-the-art solvers: `HOT` Zhang et al. (2025), `ShortCut` Schmitzer (2016), and `M3S` Chen et al. (2022). For non-grid experiments, we compare against the state-of-the-art solver `HiRef` Halmos et al. (2025) and the standard `Sinkhorn` Cuturi (2013); Flamary et al. (2021). Detailed dataset settings are provided in the Appendix E.

All experiments run on dual Intel Xeon Gold 6330 CPUs (2.0GHz), 503GB RAM, and an NVIDIA RTX 4090D (24GB). Host memory is limited to 100GB and GPU memory to 24GB (violations reported as OOM). Each instance has a 3600s wall-clock limit (violations reported as TO).

For `HALO`, we use `cuPDLPx` Lu et al. (2025) by default, while additional results demonstrating the flexibility of the LP solver choices are provided in Appendix H. For the other baselines, we use their open-source implementations with default parameter settings, with more details provided in Appendix D.

Since different methods may use different internal stopping rules, we report solution quality using unified metrics:

$$\operatorname{gap} = \frac{|\langle \boldsymbol{c}, \boldsymbol{x}\rangle - \langle \boldsymbol{c}, \boldsymbol{x}_b\rangle|}{|\langle \boldsymbol{c}, \boldsymbol{x}_b\rangle| + 1} \quad \text{and} \quad \operatorname{feas} = \max\left\{ \frac{\|\min(\boldsymbol{x}, 0)\|}{1 + \|\boldsymbol{x}\|}, \frac{\|\boldsymbol{A}\boldsymbol{x} - \boldsymbol{q}\|}{1 + \|\boldsymbol{q}\|} \right\},$$

where $\boldsymbol{x}_b$ is a high-accuracy reference obtained by solving the reduced OT model with `Gurobi` (Barrier with crossover) at a tolerance of $10^{-8}$, following Zhang et al. (2025).

### 3.1 THE IMAGE DATASET DOTMARK

For image dataset DOTmark, we use $r$ to denote the image resolution; the problem size thus scales as $n = m = r^2$. Table 1 summarizes four metrics: runtime (s), peak memory (GB), relative objective gap, and feasibility error. While providing comparable solutions in terms of gap and feas, `HALO` is memory-efficient and delivers short wall-clock time, clearly outperforming strong baselines, especially at large scales. Against `HOT`, `HALO` leads on runtime, memory, and gap. Specifically, at $r = 512$ it is 7.02× faster with 89.2% less memory, achieving 2–3 orders of magnitude tighter gaps across scales. Despite `ShortCut`'s CPU memory efficiency, `HALO` is much faster at a comparable gap, yielding a 37.36× speedup at $r = 512$. Finally, while the entropy-regularized `M3S` also solves the $r = 1024$ instances, `HALO` shows simultaneous advantages in runtime, memory, and accuracy. At $r = 1024$ it is 8.92× faster, uses 70.5% less memory, and attains 1–2 orders tighter gaps.

Figure 3 reports runtime and memory scalability on log–log axes. `HALO` shows an approximately straight runtime curve with slope $\approx 1$, indicating near-linear growth and outperforming other baselines. Its memory curve has a terminal slope $\approx 2$, matching `M3S`, yet `HALO` maintains the lowest memory usage among GPU-based methods across resolutions.

Table 1: The numerical results on DOTmark. "GPU/CPU memory" denotes GPU VRAM for GPU-based methods and CPU RAM for CPU methods (shown in gray). Time is reported in seconds (s) and memory is in gigabytes (GB). gap at $r = 512, 1024$ are unavailable because solving the reduced model with Gurobi runs out of memory.

| Metric | Resolution | 64 | 128 | 256 | 512 | 1024 |
|---|---|---|---|---|---|---|
| time | HALO | 1.50 | 2.20 | **4.31** | **11.17** | **27.73** |
| | HOT | 1.56 | **2.12** | 14.32 | 78.43 | OOM |
| | ShortCut | **0.25** | 2.41 | 25.74 | 438.14 | TO |
| | M3S | 1.95 | 3.44 | 8.51 | 39.32 | 247.22 |
| GPU/CPU memory | HALO | **0.38** | **0.48** | **0.76** | **2.07** | **6.25** |
| | HOT | 0.84 | 1.09 | 3.10 | 19.25 | OOM |
| | ShortCut | 0.01 | 0.04 | 0.14 | 0.56 | TO |
| | M3S | 0.71 | 0.95 | 1.92 | 5.78 | 21.21 |
| gap | HALO | 1.23E−6 | 1.51E−5 | 1.41E−5 | – | – |
| | HOT | 6.77E−4 | 6.03E−3 | 3.32E−2 | – | – |
| | ShortCut | 1.56E−6 | 4.04E−6 | 2.27E−5 | – | – |
| | M3S | 1.89E−4 | 1.90E−4 | 3.31E−4 | – | – |
| feas | HALO | 3.66E−8 | 2.45E−7 | 1.28E−7 | 1.06E−7 | 6.98E−8 |
| | HOT | 5.51E−7 | 7.91E−7 | 7.61E−7 | 3.42E−7 | OOM |
| | ShortCut | 1.65E−18 | 1.65E−18 | 9.55E−19 | 3.67E−19 | TO |
| | M3S | 2.40E−6 | 1.20E−6 | 6.13E−7 | 3.03E−7 | 1.44E−7 |

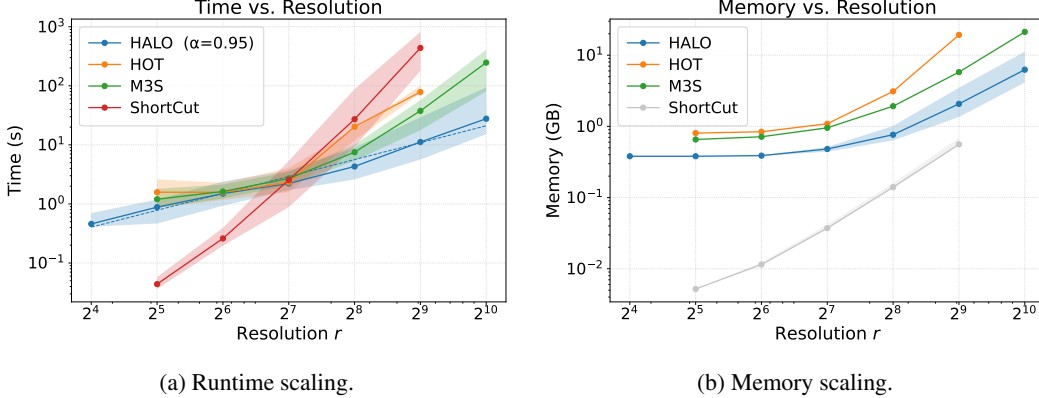

(a) Runtime scaling.         (b) Memory scaling.

Figure 3: Scalability on DOTmark across all evaluated methods. Left: Runtime scaling; Right: Memory scaling. In the left legend, $\alpha$ is the fitted slope for HALO. ShortCut uses CPU memory, shown in gray in the right panel.

To validate Theorem 1, Table 2 lists the average number of inner iterations per scale in Algorithm 2 across resolutions. The averages never exceed 2 and even tend to decrease at finer scales, corroborating the $\mathcal{O}(1)$ bound and illustrating the practical efficiency of updateActive.

Table 2: Number of inner iterations per scale.

| Scale | 16 | 32 | 64 | 128 | 256 | 512 | 1024 |
|---|---|---|---|---|---|---|---|
| scale0 | 1.00 | 1.91 | 1.82 | 1.22 | 1.12 | 1.05 | 1.00 |
| scale1 | | 1.00 | 1.80 | 1.65 | 1.15 | 1.09 | 1.05 |
| scale2 | | | 1.00 | 1.75 | 1.63 | 1.17 | 1.09 |
| scale3 | | | | 1.00 | 1.78 | 1.66 | 1.17 |
| scale4 | | | | | 1.00 | 1.77 | 1.66 |
| scale5 | | | | | | 1.00 | 1.78 |
| scale6 | | | | | | | 1.00 |

To understand the impact of data geometry on solver efficiency, we analyzed the runtime of `HALO` across different image classes in DOTmark (see Figure 4 for examples). As shown in Table 3, the runtime is highly correlated with *pixel intensity sparsity*, defined as the percentage of pixels with strictly zero mass. Classes with low pixel intensity sparsity, such as ClassicImages, converge fast, while those with high pixel intensity sparsity, such as Shapes and Microscopy, require significantly more time. This phenomenon can be theoretically explained by Theorem 1: high pixel intensity sparsity often corresponds to non-convex supports with singularities Luo et al. (2022), which implies a relatively larger Lipschitz constant $L$ in Assumption 4, resulting in a larger constant in the iteration bound and thus longer runtime.

Table 3: Performance breakdown by image class in DOTmark at resolution $1024 \times 1024$. Metric sparsity denotes the *pixel intensity sparsity*, defined as the percentage of pixels with strictly zero mass. Time is reported in seconds (s)

| Metric | WhiteNoise | GRF / Log (Avg) | ClassicImages | Cauchy | Shapes | Micro. |
|---|---|---|---|---|---|---|
| sparsity | 0.00% | 0.00% | 0.01% | 0.00% | 45.3% | 42.0% |
| time | 18.29 | 21.76 | 23.78 | 25.82 | 44.39 | 56.22 |

Table 4 presents an ablation that isolates the effects of the multiscale framework and the GPU-based LP solver `cuPDLPx`. When `cuPDLPx` is disabled, we use `Gurobi`'s barrier with crossover, as `updateActive` relies on the sparsity of solutions. Disabling `cuPDLPx` in `HALO` results in a $36.9\times$ increase in runtime at $r = 256$. Removing the multiscale framework from `HALO` also causes an $85.6\times$ slowdown at $r = 64$ and leads to OOM at higher resolutions. Taken together, the multiscale framework and `cuPDLPx` are both indispensable, yielding short wall-clock time and low memory across all tested resolutions.

Table 4: Ablation on `HALO`. An '✗' in the PDHG-based column indicates that `cuPDLPx` is replaced by `Gurobi`'s barrier method with crossover.

| Multiscale | PDHG-based | Resolution | 32 | 64 | 128 | 256 |
|---|---|---|---|---|---|---|
| ✓ | ✓ | time | **0.88** | **1.50** | **2.19** | **4.31** |
|  |  | GPU memory | 0.38 | 0.39 | 0.48 | 0.76 |
| ✓ | ✗ | time | 0.91 | 2.56 | 27.68 | 159.06 |
|  |  | CPU memory | 0.07 | 0.07 | 0.29 | 1.18 |
| ✗ | ✓ | time | 2.87 | 128.4 | OOM | OOM |
|  |  | GPU memory | 0.54 | 3.00 | OOM | OOM |
| ✗ | ✗ | time | 6.88 | 126.94 | OOM | OOM |
|  |  | CPU memory | 0.78 | 12.69 | OOM | OOM |

Table 5 further presents an ablation of the dual-violation correction in `updateActive`. We report the maximum and mean runtime over all DOTmark instances. The augmentation markedly improves robustness on difficult cases; at $r = 1024$, for instance, the maximum runtime falls to 24.3% of that without this component.

Table 6 reports the speedup of `HALO` with a constant stepsize over the power-iteration choice in `cuPDLPx`. We show results for resolutions from 256 to 1024; at $r = 1024$, the constant stepsize yields a $1.97\times$ speedup, indicating that eliminating per-iteration norm estimation substantially reduces computational cost.

## 3.2 The Point Clouds Dataset ModelNet10

To validate generalization, we evaluate `HALO` on the 3D point cloud dataset ModelNet10 (see App. C.3 for 2D results). As shown in Table 7, `HALO` demonstrates superior scalability and efficiency on non-grid domains. While `Sinkhorn` runs out of memory at $n = 2^{16}$ and `HiRef` fails at $n = 2^{19}$, `HALO` successfully scales to $n = 2^{19}$, consuming only 2.99 GB memory. Specifically at $n = 2^{18}$, `HALO` achieves a $1.84\times$ speedup and an 83.2% reduction in memory usage compared to `HiRef`. Crucially, `HALO` attains a significantly lower transport cost, improving upon `HiRef` by approximately 24.9%, demonstrating the superior precision of `HALO`.

Table 5: Ablation on `updateActive`: dual-violation augmentation (✓) improves robustness

| dual-violation | Resolution | 256 | 512 | 1024 |
|---|---|---|---|---|
| ✓ | Max time | **11.99** | **54.36** | **154.20** |
|   | Avg time | **4.31** | **11.17** | **27.73** |
| ✗ | Max time | 29.07 | 151.98 | 633.34 |
|   | Avg time | 5.47 | 15.23 | 52.85 |

Table 6: Speedup of constant step-size over power iteration.

| Resolution | 256 | 512 | 1024 |
|---|---|---|---|
| Speedup | 1.80× | 1.65× | 1.97× |

Table 7: Performance on Non-Grid 3D Data (ModelNet10). gap denotes the relative objective difference: for $n = 2^{14}$, the reference is the exact solution computed by the standard EMD solver Flamary et al. (2021); for $n \geq 2^{15}$ where EMD solver is intractable, the reference is the solution of `HALO`.

| Metric | Method | $2^{14}$ | $2^{15}$ | $2^{16}$ | $2^{17}$ | $2^{18}$ | $2^{19}$ |
|---|---|---|---|---|---|---|---|
| time | HALO | **15.54** | **26.25** | **47.42** | **88.51** | **229.7** | **444.3** |
|  | HiRef | 23.60 | 46.40 | 94.40 | 189.9 | 422.7 | OOM |
|  | Sinkhorn | 29.50 | OOM | OOM | OOM | OOM | OOM |
| memory | HALO | **0.54** | **0.64** | **0.77** | **1.12** | **1.83** | **2.99** |
|  | HiRef | 0.91 | 1.03 | 1.67 | 3.60 | 10.92 | OOM |
|  | Sinkhorn | 10.60 | OOM | OOM | OOM | OOM | OOM |
| gap | HALO | +5.33E−5 | − | − | − | − | − |
|  | HiRef | +4.19E−1 | +3.51E−1 | +3.13E−1 | +2.77E−1 | +2.49E−1 | OOM |
|  | Sinkhorn | +3.20E−2 | OOM | OOM | OOM | OOM | OOM |

## 4 DISCUSSION

In this paper, we presented `HALO`, a scalable and memory-efficient solver for large-scale optimal transport problems with squared Euclidean cost. Our work offers a key insight into large-scale OT: **by synergizing a hierarchy framework with GPU-based LP solvers, it is feasible to simultaneously achieve memory efficiency, computational speed, and high precision, without resorting to regularization or approximation methods.** Specifically, `HALO` integrates a hierarchical framework with a rigorous active-support update rule and a parallel-friendly LP solver. Furthermore, the dual-violation correction enhances robustness, a proven scale-independent iteration bound explains the fast convergence, and the Pock–Chambolle rescaling justifies the rationality of constant stepsize.

Regarding the extension to high dimensions, while the shielding-based component poses certain scaling challenges , our ongoing investigations demonstrate that the hierarchical framework possesses significant potential to effectively address high-dimensional problems through more refined active-support updating strategies.

Finally, for general transport costs (e.g., Euclidean cost), a current limitation is the reduced sparsity of solutions returned by first-order solvers compared to the squared Euclidean case, which diminishes the efficiency of the refinement phase. To address this, potential solutions include integrating future crossover algorithms to recover sparsity or designing more flexible active-support detection strategies.

## ETHICS STATEMENT

We confirm adherence to the ICLR Code of Ethics. This paper develops a hierarchical algorithm for large-scale optimal transport and does not involve human subjects, clinical data, or personally identifiable information. All datasets used are publicly available under their respective licenses; we applied only standard preprocessing and did not attempt re-identification or attribute inference. The method is general-purpose, so any fairness concerns stem from downstream data and deployment contexts; practitioners should audit subgroup performance in their applications. To reduce misuse risks, we restrict experiments to benign public datasets and will document intended use and limitations. Experiments were run on local institutional hardware without external data services; compute configuration and runtimes are reported in the appendix to encourage reuse and minimize redundant computation. To the best of our knowledge, this work complies with applicable laws and institutional policies; no IRB approval was required. The authors declare no conflicts of interest and no third-party sponsorship that could unduly influence the results.

## REPRODUCIBILITY STATEMENT

Complete proofs are provided in the appendix: Theorem 1 is proved in Appendix B.1, and Proposition 1 is proved in Appendix B.2. Implementation details for all baselines are given in Appendix D. Dataset-related details are provided in Appendix E.

## ACKNOWLEDGEMENTS

This work was supported by the NSFC (grant no. 12090024), the Natural Science Foundation of Chongqing, China (CSTB2023NSCQ-LZX0054).

## CODE AVAILABILITY

Our implementation of `HALO` is available at `https://github.com/WenzhouXia/HALO`.

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

## A  THE USE OF LARGE LANGUAGE MODELS (LLMs)

We used a large language model solely to polish writing and to assist with small code snippets. It did not generate ideas, proofs, experimental designs, or results. All LLM-assisted content was authored, reviewed, and validated by the authors.

## B  PROOFS OF THEORETICAL RESULTS

### B.1  PROOF OF THEOREM 1

**Proof 1 (Proof of Theorem 1)** *Fix $s \in \mathbb{S}$ and let $\widehat{D}_{k+1}(s) \subset \mathbb{D}$ denote the set of targets $d$ that are not shielded from $s$ by the pairs $\{(s', \mathrm{map}_k(s')) : s' \in S(s)\}$. Hence every $(s, d)$ with $d \in \widehat{D}_{k+1}(s)$ must be explicitly added to $\mathbb{N}_{k+1}$.*

*We first establish a geometric localization bound. Take $d \in \widehat{D}_{k+1}(s)$ and set $v = d - \mathrm{map}_k(s)$. By assumption 1 there exists $s' \in S(s)$ such that*

$$\langle v,\, s' - s \rangle \ \geq \ \|v\|\, \|s' - s\|\, q.$$

*Insert and subtract $\mathrm{map}_k(s')$ and apply assumption 4 together with Cauchy–Schwarz to obtain*

$$\begin{aligned}
\langle v,\, s' - s \rangle &= \langle d - \mathrm{map}_k(s) + \mathrm{map}_k(s) - \mathrm{map}_k(s'),\, s' - s \rangle \\
&\geq \|d - \mathrm{map}_k(s)\|\, \|s' - s\|\, q - \|\mathrm{map}_k(s) - \mathrm{map}_k(s')\|\, \|s' - s\| \\
&\geq \big(q\, \|d - \mathrm{map}_k(s)\| - L\|s' - s\|\big)\, \|s' - s\|.
\end{aligned}$$

*By assumption 2 we have $\|s' - s\| < D$, hence*

$$\langle d - \mathrm{map}_k(s),\, s' - s \rangle \ \geq \ \big(q\, \|d - \mathrm{map}_k(s)\| - LD\big)\, \|s' - s\|.$$

*If $\|d - \mathrm{map}_k(s)\| > LD/q$, then $\langle d - \mathrm{map}_k(s), s' - s \rangle > 0$ for some $s' \in S(s)$. For the squared Euclidean ground cost this is exactly the sufficient shielding condition stated in Section 5.2 of Schmitzer (2016). Therefore each unshielded $d$ must satisfy*

$$\|d - \mathrm{map}_k(s)\| \leq R := \frac{LD}{q}, \qquad \text{that is,} \qquad \widehat{D}_{k+1}(s) \subset B_R(\mathrm{map}_k(s)).$$

*Assumption 5 gives $\|\mathrm{map}_k(s) - \mathrm{map}_0(s)\| \leq R_0$ for all $k$, which implies*

$$\widehat{D}_{k+1}(s) \subset B_{R_0+R}(\mathrm{map}_0(s)) \quad \text{for all } k \geq 0.$$

*Taking the union over all iterations,*

$$\bigcup_{k \geq 0} \widehat{D}_{k+1}(s) \subset \mathbb{D} \cap B_{R_0+R}(\mathrm{map}_0(s)).$$

*By assumption 3 the cardinality is uniformly bounded as*

$$\left| \bigcup_{k \geq 0} \widehat{D}_{k+1}(s) \right| \leq \left| \mathbb{D} \cap B_{R_0+R}(\mathrm{map}_0(s)) \right| \leq \rho \, \mathrm{vol}_n(B_{R_0+R}) := C_0,$$

*which depends only on the constants in assumptions 1–5, on the dimension of the space, and is independent of the sizes of $\mathbb{S}$ and $\mathbb{D}$.*

*We now conclude the proof. Since each $s$ has at most $C_0$ unshielded targets, if at iteration $k$ there is still some $d \in \widehat{D}_{k+1}(s)$, then at least one new pair $(s, d)$ must be added. As the total number of such unshielded pairs is bounded by $C_0$, the process of adding new shielding edges can occur at most $C_0$ times. Thus the sequence $(\mathbb{N}_k)$ converges after at most $C_0$ iterations, which proves scale-independent convergence.*

*Regarding sparsity, the shielding part contributes at most $C_0|\mathbb{S}|$ pairs. Step 3 of Algorithm 3 can add at most $K$ pairs per iteration, and since there are at most $C_0$ iterations, its total contribution is bounded by $KC_0$. Therefore the final support satisfies*

$$|\mathbb{N}_k| \leq C_0|\mathbb{S}| + \beta C_0|\mathbb{S}| = (1 + \beta)C_0|\mathbb{S}|.$$

*Thus we can take $C := (1 + \beta)C_0$, which depends only on the constants in assumptions 1–5, on the space dimension, and on the chosen parameter $\beta$. At the fixed point $\boldsymbol{x}_k$ is locally optimal with respect to $\mathbb{N}_k$, and by construction $\mathbb{N}_k$ is a shielding neighbourhood. By the local-to-global certification (Corollary 3.10 of Schmitzer (2016)), $\boldsymbol{x}_k$ is globally optimal once the process stabilizes. This completes the proof.*

## B.2 PROOF OF PROPOSITION 1

**Proof 2** *We first compute $\boldsymbol{B}\boldsymbol{B}^\top$:*

$$\boldsymbol{B}\boldsymbol{B}^\top = \begin{bmatrix} diag(r) & \boldsymbol{P} \\ \boldsymbol{P}^\top & diag(c) \end{bmatrix},$$

*where $\boldsymbol{P} \in \{0,1\}^{m \times n}$ is the adjacency matrix of a bipartite graph representing the non-zero entries in $\boldsymbol{B}$. Next, we compute the rescaled matrix $\widetilde{\boldsymbol{B}}\widetilde{\boldsymbol{B}}^\top$:*

$$\widetilde{\boldsymbol{B}}\widetilde{\boldsymbol{B}}^\top = \frac{1}{2}D_r^{-1}(\boldsymbol{B}\boldsymbol{B}^\top)D_r^{-1} = \frac{1}{2}\begin{bmatrix} \boldsymbol{I}_m & \boldsymbol{Q} \\ \boldsymbol{Q}^\top & \boldsymbol{I}_n \end{bmatrix}, \quad \boldsymbol{Q} := D_r^{-1/2}\boldsymbol{P}D_c^{-1/2},$$

*where $D_r = diag(r)$ and $D_c = diag(c)$.*

*For the matrix $\boldsymbol{Q}$, it is known that the eigenvalues of the block matrix $\begin{bmatrix} \boldsymbol{I} & \boldsymbol{Q} \\ \boldsymbol{Q}^\top & \boldsymbol{I} \end{bmatrix}$ are $1 \pm \sigma_i(\boldsymbol{Q})$, thus:*

$$\lambda_{\max}(\widetilde{\boldsymbol{B}}\widetilde{\boldsymbol{B}}^\top) = \frac{1}{2}(1 + \sigma_{\max}(\boldsymbol{Q})), \quad \|\widetilde{\boldsymbol{B}}\|_2 = \sqrt{\frac{1 + \sigma_{\max}(\boldsymbol{Q})}{2}}.$$

*We now prove that $\sigma_{\max}(\boldsymbol{Q}) = 1$.*

***Lower bound:*** *Let* $\boldsymbol{u} = (\sqrt{r_1}, \ldots, \sqrt{r_m})^\top$ *and* $\boldsymbol{v} = (\sqrt{c_1}, \ldots, \sqrt{c_n})^\top$. *We have:*

$$(\boldsymbol{Q}\boldsymbol{v})_i = \sum_{j \in N(i)} \frac{\sqrt{c_j}}{\sqrt{r_i c_j}} = \frac{r_i}{\sqrt{r_i}} = \sqrt{r_i} = u_i.$$

*Similarly,* $\boldsymbol{Q}^\top \boldsymbol{u} = \boldsymbol{v}$, *so* $\boldsymbol{u}$ *and* $\boldsymbol{v}$ *are singular vectors, and the singular value is exactly 1. Thus,* $\sigma_{\max}(\boldsymbol{Q}) \geq 1$.

***Upper bound:*** *For any* $\boldsymbol{x} \in \mathbb{R}^n$, *by the Cauchy-Schwarz inequality:*

$$\|\boldsymbol{Q}\boldsymbol{x}\|_2^2 = \sum_i \frac{1}{r_i} \left( \sum_{j \in N(i)} \frac{x_j}{\sqrt{c_j}} \right)^2 \leq \sum_i \frac{1}{r_i} \left( \sum_{j \in N(i)} 1 \right) \left( \sum_{j \in N(i)} \frac{x_j^2}{c_j} \right) = \sum_i \sum_{j \in N(i)} \frac{x_j^2}{c_j} = \sum_j x_j^2.$$

*Here,* $N(i)$ *denotes the indices of the non-zero elements in the adjacency matrix* $\boldsymbol{P}$. *Thus,* $\|\boldsymbol{Q}\|_2 \leq 1$.

*Therefore,* $\sigma_{\max}(\boldsymbol{Q}) = 1$, *and we conclude:*

$$\|\widetilde{\boldsymbol{B}}\|_2 = \sqrt{\frac{1+1}{2}} = 1.$$

## C  EXTENSION TO NON-GRID DATA

### C.1  CONSTRUCTION OF HIERARCHICAL STRUCTURE

For non-grid data (e.g., point clouds), we primarily employ a $2^d$-tree structure (e.g., Quadtrees in 2D, Octrees in 3D) to construct the hierarchy via spatial partitioning. The procedure starts by defining an axis-aligned bounding hypercube containing all data points, which serves as the coarsest level. Finer levels are generated by subdividing each hypercube into $2^d$ equal-sized sub-cubes, discarding any sub-cubes that contain no data. This subdivision continues recursively until a pre-determined depth is reached or the finest level (containing individual points) is achieved. At each level, the active nodes correspond to these non-empty sub-cubes, and the representative point is defined as the geometric center of the spatial region.

In higher dimensions, standard $2^d$-trees become intractable due to the $2^d$ branching factor. To address this, sequential axis partitioning (similar to **k-d trees**) can be adopted to control the inter-level reduction ratio, ensuring hierarchy construction does not become a computational bottleneck.

### C.2  SHIELDING STRATEGY ON NON-GRID DATA

In the shielding-based active support update (Algorithm 3), the definitions of the local neighborhood $\mathcal{R}(s)$ and the unshielded set $\hat{D}(s)$ require adaptation for non-grid domains. In the image setting (2D grids), $\mathcal{R}(s)$ consists of the 8 surrounding pixels. To maintain consistency and ensure extensibility to high dimensions, we replace this with K-Nearest Neighbors (KNN), setting $k_{\text{nn}} = 4 \times d$ (e.g., $k_{\text{nn}} = 12$ for 3D point clouds). This choice aligns with the image setting (where $8 = 4 \times 2$) and keeps the neighborhood sparse.

Unlike grids where the spatial distribution is uniform, non-grid data exhibit irregular distributions. Consequently, a single source point may correspond to a large number of unshielded candidates. For computational and memory efficiency, we enforce an upper bound on the number of candidates added per iteration, denoted as $U_{\max}$.

Crucially, this budgeted strategy affects neither the convergence of `HALO` nor the theoretical result of Theorem 1. Regarding convergence, hierarchical algorithms rely on the monotonic decrease of the objective function, which is guaranteed as long as $\mathbb{N}_{k+1} \supset \text{supp}(\boldsymbol{x}_k)$. Regarding scale-independence, the complexity bound in Theorem 1 is derived from the total volume of potential unshielded targets (bounded by the constant $C_0$ in the proof B.1). While the theorem assumes all unshielded targets are identified, limiting the update size to $U_{\max}$ merely implies that these necessary neighbors are added over slightly more iterations, preserving the scale-independent complexity.

Finally, we provide a sensitivity analysis in Table 8 and Table 9 to demonstrate the stability of our algorithm with respect to this hyperparameter. We tested $U_{\max} \in \{10, 15, 20, 30, 40\}$ on ModelNet10

(see Appendix E). The results indicate that the performance remains stable across varying scales, and we use $U_{\max} = 20$ as a safe default setting, which is used throughout non-grid experiments.

Table 8: Time (s) ablation on the budget size $U_{\max}$ for shielding on 3D point clouds.

| $U_{\max}$ | $2^{14}$ | $2^{15}$ | $2^{16}$ | $2^{17}$ | $2^{18}$ |
|---|---|---|---|---|---|
| 10 | 16.18 | 28.32 | 50.62 | 92.82 | 265.8 |
| 15 | 15.70 | 26.83 | 48.92 | **88.15** | 236.6 |
| 20 | **15.54** | 26.25 | **47.42** | 88.51 | 229.7 |
| 30 | 15.85 | 27.58 | 50.18 | 91.30 | **224.7** |
| 40 | 16.04 | **26.09** | 51.81 | 103.7 | 244.2 |

Table 9: Memory (GB) ablation on the budget size $U_{\max}$ for shielding on 3D point clouds.

| $U_{\max}$ | $2^{14}$ | $2^{15}$ | $2^{16}$ | $2^{17}$ | $2^{18}$ |
|---|---|---|---|---|---|
| 10 | **0.54** | 0.62 | **0.74** | **1.06** | **1.72** |
| 15 | 0.54 | 0.63 | 0.76 | 1.10 | 1.76 |
| 20 | 0.54 | 0.64 | 0.77 | 1.12 | 1.83 |
| 30 | 0.55 | 0.65 | 0.79 | 1.17 | 1.92 |
| 40 | 0.56 | 0.61 | 0.83 | 1.21 | 1.95 |

### C.3 NUMERICAL RESULTS ON 2D NON-GRID DATA

In addition to the 3D experiments presented in Table 7, we provide results on 2D non-grid data constructed from ModelNet10 (see Appendix E for details). The results are summarized in Table 10. Similar to the 3D setting, HALO demonstrates consistent superior performance. At $n = 2^{18}$, it achieves a $3.47\times$ speedup, an $82.4\%$ reduction in memory usage compared to HiRef, and a $32.5\%$ lower transport cost.

Table 10: Performance on 2D Non-Grid Data (ModelNet10-PCA). gap denotes the relative objective difference: for $n = 2^{14}$, the reference is the exact solution computed by the standard EMD solver Flamary et al. (2021); for $n \geq 2^{15}$ where EMD solver is intractable, the reference is the solution of HALO.

| Metric | Method | $2^{14}$ | $2^{15}$ | $2^{16}$ | $2^{17}$ | $2^{18}$ | $2^{19}$ |
|---|---|---|---|---|---|---|---|
| | HALO | **7.82** | **12.50** | **23.44** | **53.25** | **121.6** | **247.0** |
| time | HiRef | 23.40 | 46.40 | 94.10 | 190.4 | 422.0 | OOM |
| | Sinkhorn | 28.50 | OOM | OOM | OOM | OOM | OOM |
| | HALO | **0.55** | **0.62** | **0.74** | **1.08** | **1.92** | **3.97** |
| memory | HiRef | 0.90 | 1.02 | 1.65 | 3.60 | 10.89 | OOM |
| | Sinkhorn | 10.60 | OOM | OOM | OOM | OOM | OOM |
| | HALO | +5.68E−5 | − | − | − | − | − |
| gap | HiRef | +4.37E−1 | +3.82E−1 | +4.04E−1 | +3.00E−1 | +3.25E−1 | OOM |
| | Sinkhorn | +4.02E−2 | OOM | OOM | OOM | OOM | OOM |

## D ALGORITHM SETTINGS

For HALO, we employ the cuPDLPx Lu et al. (2025) solver with constant step-sizes 1 unless otherwise specified. All experiments use the Pock–Chambolle rescaling scheme, and the stopping criterion is set uniformly with primal feasibility, dual feasibility, and objective gap thresholds of $10^{-6}$. In updateActive, we choose $K = 0.25|\mathbb{S}|$ for the operator $\text{Top}_K$

For HOT Zhang et al. (2025), we adopt the open-source implementation with its default parameter choices. The stopping criterion is fixed to $10^{-6}$ to ensure comparability with other baselines.

For ShortCut Schmitzer (2016), we use the variant based on the LEMON solver provided in the released code, with all default parameters left unchanged.

For M3S Chen et al. (2022), which is an entropic regularization method, we use the official implementation with its predefined entropy-regularization coefficient and all other default parameter settings.

For Gurobi Gurobi Optimization, LLC (2024), we rely on the Barrier algorithm. Unless otherwise specified in the main text, the crossover procedure is disabled.

For `HiRef` Halmos et al. (2025), we adopt the official open-source implementation and strictly adhere to the default parameter settings provided by the authors.

For `Sinkhorn`, we employ the standard implementation provided by the POT library Flamary et al. (2021). We set the regularization parameter to $\varepsilon = 10^{-3}$ to obtain an approximate solution with reasonable accuracy.

## E  DATASET

To comprehensively evaluate the scalability and generalization of `HALO`, we employed two distinct datasets covering both grid-based images and unstructured point clouds.

**DOTmark (Image Data).**  To evaluate scalability across different problem sizes on grid data, we utilized the DOTmark Schrieber et al. (2016) benchmark. In addition to the native resolutions (32 to 512), we constructed additional variants by both downsampling and upsampling. For downsampling, each native-resolution image was resized to $16 \times 16$ using bilinear interpolation. The interpolated values were rescaled linearly to match the original intensity range, rounded to integers, and clipped to avoid numerical overflow. For upsampling, we generated $1024 \times 1024$ images by bilinear interpolation, followed by rounding to integers. To guarantee exact consistency with the original data, each pixel at the original grid was enforced to coincide with its corresponding position in the enlarged image, and the resulting values were clipped to the original intensity range before being stored. In this way, the $16 \times 16$ images provide small-scale test cases, while the $1024 \times 1024$ images serve as challenging large-scale benchmarks.

**ModelNet10 (Non-Grid Data).**  To validate the performance of `HALO` on non-grid data, we constructed a benchmark using ModelNet10 Wu et al. (2015), a widely used dataset for 3D point cloud analysis. We selected the top-3 samples from each of the 10 classes, generating a total of 30 pairs of point clouds for evaluation. The raw point clouds were normalized to the unit hypercube, and we varied the number of points $n$ from $2^{11}$ to $2^{19}$ via random sampling to test scalability. To further evaluate the algorithm on non-grid 2D data, we generated 2D counterparts of these 3D shapes using Principal Component Analysis (PCA), creating a non-grid 2D point cloud benchmark distinct from the regular grids in DOTmark.

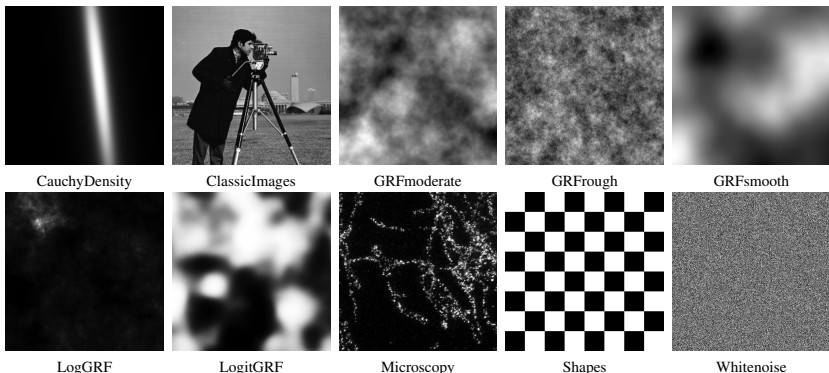

Figure 4: Example images from the DOTmark benchmark.

## F  SENSITIVITY ANALYSIS OF HYPERPARAMETER IN DUAL-VIOLATION CORRECTION

A key hyperparameter in `HALO` is $\beta$, associated with the dual-violation correction step in Algorithm 3. It controls the size of the candidate set added during the update by selecting the top $K = \beta|\mathbb{S}|$ pairs with the largest dual violations. To assess the sensitivity of `HALO` to this parameter, we evaluated the algorithm on the DOTmark benchmark across a wide range of values:

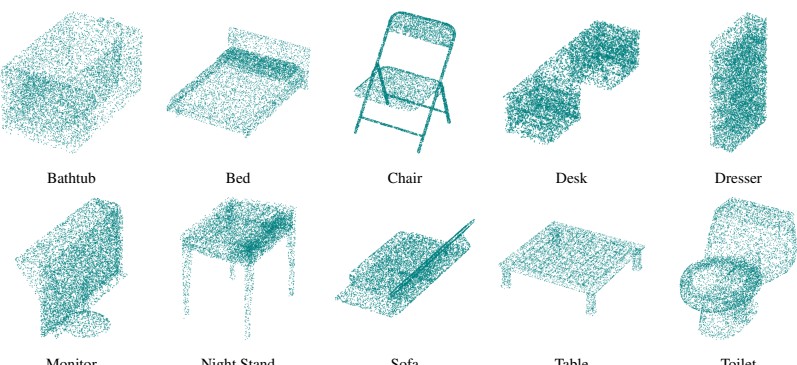

Figure 5: Example 3D point clouds from the ModelNet10 benchmark. Each object is sampled with $n = 8192$ points.

$\beta \in \{2^{-4}, 2^{-3}, 2^{-2}, 2^{-1}, 2^0\}$. We also included the baseline case $\beta = 0$, which actually disables the dual-violation correction module.

The numerical results are detailed in Table 11. The algorithm exhibits high stability regarding both memory and runtime: GPU memory usage remains virtually unaffected by the choice of $\beta$; runtime also shows minimal fluctuation, with a maximum variation of approximately 17% at resolution $1024 \times 1024$. However, setting $\beta = 0$ leads to a sharp performance drop, which confirms that the dual-violation correction is crucial for the efficiency of the solver.

Based on these findings, we recommend $\beta = 2^{-2}$ as a safe default setting, which is used throughout our main experiments.

Table 11: Sensitivity analysis of $\beta$ on DOTmark. Performance remains stable for $\beta \in [2^{-3}, 2^0]$. Setting $\beta = 0$ leads to significant slowdowns. Memory usage is consistent across all non-zero $\beta$ values.

|  | $\beta$ | $r = 64$ | $r = 128$ | $r = 256$ | $r = 512$ | $r = 1024$ |
|---|---|---|---|---|---|---|
| | $2^0$ | 0.77 | 1.18 | 2.79 | 9.24 | 28.86 |
| | $2^{-1}$ | 0.78 | 1.12 | 2.70 | 8.76 | 25.19 |
| | $2^{-2}$ | 0.78 | **1.11** | **2.62** | 8.84 | 25.42 |
| time | $2^{-3}$ | **0.72** | **1.11** | 2.71 | **8.72** | **24.67** |
| | $2^{-4}$ | 1.73 | 2.42 | 4.44 | 11.90 | 29.64 |
| | 0 (No dual) | 1.90 | 2.99 | 5.47 | 15.23 | 52.85 |
| memory | All | $\sim 0.39$ | $\sim 0.48$ | $\sim 0.76$ | $\sim 2.10$ | $\sim 6.30$ |

## G   COMPARISON WITH A GPU-BASED SHORTCUT IMPLEMENTATION

A natural question arises regarding whether the `ShortCut` method could yield even better results than `HALO` if implemented on GPUs. To clarify this, we implemented a `ShortCut-GPU` variant within the same framework as `HALO`, utilizing `cuPDLPx` as the underlying solver. The primary difference lies in the active-support update strategy: `ShortCut` employs an aggressive pruning strategy that retains only the support of the current coupling, whereas `HALO` uses a conservative update ($\mathbb{N}_{k+1} \supset \mathbb{N}_k$) augmented with dual-violation correction.

The comparison results on DOTmark are presented in Table 12. While the GPU implementation brings efficiency gains to `ShortCut-GPU` compared to CPU baselines, `HALO` still outperforms `ShortCut-GPU` significantly, achieving a 2.5× speedup at resolution $1024 \times 1024$.

This performance gap highlights a critical algorithmic contribution of `HALO`. GPU-based first-order solvers typically yield solutions with lower precision compared to CPU-based classical methods. `ShortCut`'s aggressive active-support update is highly sensitive to this numerical noise, which

leads to stagnation during the refinement process. In contrast, `HALO`'s conservative update rule and the dual-violation correction provide the necessary stability. These designs make the hierarchical framework robust to the lower precision of GPU solvers, thereby unlocking the full potential of GPU acceleration.

Table 12: Runtime comparison between `ShortCut-GPU` and `HALO` on DOTmark. Time is in seconds (s).

| Resolution | 512 | 1024 |
|---|---|---|
| ShortCut-GPU | 21.40 | 68.33 |
| HALO | **11.17** | **27.73** |

## H    FLEXIBILITY WITH ALTERNATIVE SOLVERS

While `HALO` utilizes `cuPDLPx` Lu et al. (2025) as the default LP solver due to its state-of-the-art performance, the proposed hierarchical framework is designed to be flexible and compatible with various first-order GPU solvers. To validate this flexibility, we integrated an alternative solver, `HPR-LP` Chen et al. (2024), into `HALO` by replacing the backend of the `solveRestricted` component. The numerical results on DOTmark are reported in Table 13.

Although `HPR-LP` is generally slower than `cuPDLPx` in this context, the `HALO` framework still effectively leverages it to solve large-scale instances efficiently. Specifically, at resolution $1024 \times 1024$, `HALO` integrated with `HPR-LP` achieves a $6.1\times$ speedup and a $68.8\%$ reduction in memory usage compared to the state-of-the-art solver `M3S` (see Table 1).

This experiment confirms that the efficiency of `HALO` is not solely dependent on a specific underlying LP solver; rather, the hierarchical active-support framework is a critical component that significantly contributes to the overall performance. It is worth noting that GPU-based LP solvers are a burgeoning field compared to mature CPU solvers. We believe that the continuous evolution of faster GPU-based solvers will further boost `HALO`'s performance, solidifying our framework as a highly promising direction for solving large-scale OT.

Table 13: Performance of `HALO` integrated with the alternative `HPR-LP` solver Chen et al. (2024) on DOTmark. Time is in seconds (s) and Memory is in gigabytes (GB).

| Resolution | 64 | 128 | 256 | 512 | 1024 |
|---|---|---|---|---|---|
| time | 1.06 | 1.70 | 3.31 | 9.68 | 40.76 |
| memory | 0.72 | 0.78 | 1.00 | 1.90 | 6.61 |
| gap | 6.39E−6 | 2.36E−5 | 1.76E−5 | – | – |
| infeas | 4.09E−7 | 2.53E−7 | 1.33E−7 | 1.03E−7 | 6.08E−8 |

## I    ROBUSTNESS AND GENERALIZATION ACROSS DIVERSE DATASETS

To demonstrate generalization beyond DOTmark, we conducted additional evaluations on two high-resolution datasets: Flickr-Faces-HQ (FFHQ) Karras et al. (2019) and WikiArt Tan et al. (2019). Specifically, we selected the first 10 images from FFHQ and the first image from each of the top-10 classes in WikiArt. For each dataset, we generated 45 instances by pairing these images, strictly following the DOTmark preprocessing pipeline.

The numerical results are summarized in Table 14. `HALO` consistently achieves low memory usage and fast solving speeds on these diverse tasks, exhibiting performance metrics highly consistent with those on DOTmark. This confirms the broad applicability and robustness of `HALO` across diverse real-world datasets.

Table 14: Generalization performance of `HALO` on real-world datasets (FFHQ and WikiArt) compared to the standard DOTmark. Results are averaged over instances. Time is in seconds (s) and Memory is in gigabytes (GB).

|        | Resolution | 256  | 512   | 1024  |
|--------|------------|------|-------|-------|
|        | FFHQ       | 3.24 | 10.79 | 23.68 |
| time   | WikiArt    | 2.74 | 9.13  | 22.14 |
|        | DOTmark    | 4.31 | 11.17 | 27.73 |
|        | FFHQ       | 0.77 | 2.19  | 6.52  |
| memory | WikiArt    | 0.74 | 1.97  | 6.10  |
|        | DOTmark    | 0.76 | 2.07  | 6.25  |

