# OpenReview forum: "A Memory-Efficient Hierarchical Algorithm for Large-scale Optimal Transport Problems"
_ICLR.cc/2026/Conference — ICLR 2026 Poster_

### Official Review · Reviewer_Tm5V · 2025-10-30

**Soundness:** 4
**Presentation:** 3
**Contribution:** 3
**Rating:** 6
**Confidence:** 3

**Summary:**

This paper introduces HALO, a hierarchical and memory-efficient algorithm for large-scale optimal
transport (OT) problems on 2D supports with squared Euclidean cost. The method combines a coarseto-
fine multi-scale framework with a GPU-optimized PDHG (Primal-Dual Hybrid Gradient) solver
and an active-pruning mechanism for sparsity. The authors prove a scale-independent iteration
complexity bound for the refinement phase and show strong empirical results on the DOTmark
benchmark, achieving significant speedup and memory usage saving compared to methods such as
HOT, ShortCut, and M3S.

**Strengths:**

1. Theorem 1 (scale-independent iteration complexity) provides a nontrivial and interpretable
convergence guarantee, addressing the gap in multi-scale OT methods (which often lack formal
complexity bounds).
2. HALO merges the hierarchical multi-scale structure with a GPU-based LP solver. This is a
thoughtful contribution, given the growing interest in efficient OT solvers.
3. The paper is clearly written and well-organized. Figures 1–3 effectively illustrate the hierarchical
process and empirical trends. Proofs are included in the appendix and seem technically sound.

**Weaknesses:**

1. HALO is currently limited to 2D support with squared Euclidean cost. The hierarchical design
heavily relies on the specific problem (OT between two 2D images), while applications of OT
(like in generative models) are on more general settings (Wasserstein GAN [1] and optimal flow
matching [2]) where the supports are high-dimensional. In this case, the HALO may not work well
as in the image problems in DOTmark because the data locality property doesn't hold in these
problem. The limited problem setting is mentioned but not stated clearly in the introduction.

2. Although the paper fairly acknowledges this, HALO’s performance heavily relies on a third-party
GPU solver. A deeper analysis of how HALO interacts with other first-order solvers would make
the contribution more self-contained.

3. The parameter beta in the dual-violation step appears to influence sparsity and runtime, but the
sensitivity analyses are missing. Reporting how beta impacts convergence and memory would be
valuable.

4. In DOTMark [3], there are various types of images. It would be interesting to see how HALO
performs on these classes of images respectively to have a clearer idea of the practicality of the
assumptions and the influence of the data locality on the convergence of HALO.

References:
[1] Adler, Jonas, and Sebastian Lunz. "Banach wasserstein gan." Advances in neural information
processing systems 31 (2018).
[2] Kornilov, Nikita, et al. "Optimal flow matching: Learning straight trajectories in just one
step." Advances in Neural Information Processing Systems 37 (2024): 104180-104204.
[3] Schrieber, Jörn, Dominic Schuhmacher, and Carsten Gottschlich. "Dotmark–a benchmark for
discrete optimal transport." IEEE Access 5 (2016): 271-282.

**Questions:**

1. HALO is currently limited to 2D support with squared Euclidean cost. The hierarchical design
heavily relies on the specific problem (OT between two 2D images), while applications of OT
(like in generative models) are on more general settings (Wasserstein GAN [1] and optimal flow
matching [2]) where the supports are high-dimensional. In this case, the HALO may not work well
as in the image problems in DOTmark because the data locality property doesn't hold in these
problem. The limited problem setting is mentioned but not stated clearly in the introduction.

2. Although the paper fairly acknowledges this, HALO’s performance heavily relies on a third-party
GPU solver. A deeper analysis of how HALO interacts with other first-order solvers would make
the contribution more self-contained.

3. The parameter beta in the dual-violation step appears to influence sparsity and runtime, but the
sensitivity analyses are missing. Reporting how beta impacts convergence and memory would be
valuable.

4. In DOTMark [3], there are various types of images. It would be interesting to see how HALO
performs on these classes of images respectively to have a clearer idea of the practicality of the
assumptions and the influence of the data locality on the convergence of HALO.

References:
[1] Adler, Jonas, and Sebastian Lunz. "Banach wasserstein gan." Advances in neural information
processing systems 31 (2018).
[2] Kornilov, Nikita, et al. "Optimal flow matching: Learning straight trajectories in just one
step." Advances in Neural Information Processing Systems 37 (2024): 104180-104204.
[3] Schrieber, Jörn, Dominic Schuhmacher, and Carsten Gottschlich. "Dotmark–a benchmark for
discrete optimal transport." IEEE Access 5 (2016): 271-282.

---

> ### Author Response · Authors · 2025-11-21
>
> We thank the reviewer for the constructive feedback and positive assessment. We address the reviewer's specific concerns point-by-point below, and highlight the corresponding revisions in **red** and **orange** in the revised manuscript.
>
> ------
>
> > HALO is currently limited to 2D support with squared Euclidean cost. The hierarchical design heavily relies on the specific problem (OT between two 2D images), while applications of OT (like in generative models) are on more general settings (Wasserstein GAN [1] and optimal flow matching [2]) where the supports are high-dimensional. In this case, the HALO may not work well as in the image problems in DOTmark because the data locality property doesn't hold in these problem. The limited problem setting is mentioned but not stated clearly in the introduction.
>
> We have addressed the Extension in the **General Response 1-3**. Specifically:
>
> 1. **Beyond 2D Images :** As clarified in **Part I**, HALO can extend naturally to non-grid data like **3D point clouds**, with new experiments (**General Response, Table R1** and **Table R2**, also included in **revised Section 3.2 and Appendix C**) demonstrating HALO's superior efficiency in such settings.
>
> 2. **Application:** Consequently, HALO is directly applicable to generative tasks like **3D point cloud generation**[4]
>
> 3. **High-Dimensional Scalability (Section 4):** While we acknowledge the challenge in very high dimensions, our preliminary research confirms the hierarchical framework efficiently solves **exact OT with millions of points in >1024 dimensions on a single GPU,** as detailed in the **General Response (2. Extension to Higher Dimensions)**
>
> 4. Action: We have revised the Introduction to explicitly mention the applications of OT in generative models and clarify the current scope.
>
>    1. Revision in Application
>
>       > It has been widely adopted in various fields such as **generative modeling ([1-2])**, color transfer …
>
>    2. Revision in Scope Definition
>
>       > Our numerical results on problems with hierarchical structure, e.g. 2D images and 3D point clouds, demonstrate that our framework is efficient for computational acceleration and memory reduction.
>
> ------
>
> > Although the paper fairly acknowledges this, HALO’s performance heavily relies on a third-party GPU solver. A deeper analysis of how HALO interacts with other first-order solvers would make the contribution more self-contained.
>
> We validated HALO's flexibility by replacing cuPDLP with HPR-LP [4] (**Table R6, Appendix H)**. Even then, HALO consistently outperforms M3S (6.1$\times$ speedup, 68.8% less memory at $r=1024$). This confirms the framework itself drives efficiency, rather than relying solely on a specific LP solver, ensuring HALO naturally inherits future GPU solver advancements.
>
> **Table R6:** Performance of HALO with HPR-LP.
>
> | **Metric**   | **64**  | **128** | **256** | **512** | **1024** |
> | ------------ | ------- | ------- | ------- | ------- | -------- |
> | **Time (s)** | 1.06    | 1.70    | 3.31    | 9.68    | 40.76    |
> | **Mem (GB)** | 0.72    | 0.78    | 1.00    | 1.90    | 6.61     |
> | **Gap**      | 6.39e-6 | 2.36e-5 | 1.76e-5 | -       | -        |
> | **Infeas**   | 4.09e-7 | 2.53e-7 | 1.33e-7 | 1.03e-7 | 6.08e-8  |
>
> ------

---

> ### Author Response · Authors · 2025-11-27
>
> > The parameter beta in the dual-violation step appears to influence sparsity and runtime, but the sensitivity analyses are missing. Reporting how beta impacts convergence and memory would be valuable.
>
> We address the parameter sensitivity in **General Response (4. Sensitivity Analysis of $\beta$, Appendix F)**. By evaluating $\beta$ across a broad range ($\beta=2^{-4,-3,-2,-1,0}$ and $\beta=0$, see **Table R3**), The results confirm that the dual-violation correction is both essential and robust to parameter choice.
>
> ------
>
> > In DOTMark [3], there are various types of images. It would be interesting to see how HALO performs on these classes of images respectively to have a clearer idea of the practicality of the assumptions and the influence of the data locality on the convergence of HALO.
>
> To investigate  this insightful question, we analyzed the **pixel intensity sparsity**j, defined as percentage of zero-mass pixels, and runtime for each class in DOTmark (**Section 3.1**). As shown in **Table R7**, HALO performs consistently fast on most classes (e.g., `WhiteNoise`, ~$18.3$s), but requires more time on `Microscopy` ($56.2$s) and `Shapes` ($44.4$s). These results align with our theoretical **Assumption 4** and the insight raised by Reviewer u6SB:
>
> - Convex and High Locality (Fast)**:** `WhiteNoise` is a representative class where the $\mathrm{map}_k$ is smooth (small Lipschitz constant $L$). The geometric shielding works efficiently here, leading to the fastest convergence.
> - Non-Convex and Low Locality (Slow): `Shapes` and `Microscopy` exhibit **high pixel intensity sparsity**, forming non-convex supports. As noted by Reviewer u6SB, such geometry induces **singularities**, which implies a **relatively large Lipschitz constant $L$**[5] in the context of Assumption 4, consequently resulting in longer runtimes.
>
> **Table R7**: Performance Breakdown by Image Class ($r=1024$).
>
> | **Metric**   | **WhiteNoise** | **Others (Avg)∗** | **Shapes** | **Microscopy** |
> | ------------ | -------------- | ----------------- | ---------- | -------------- |
> | **Sparsity** | 0.00%          | ~0.00%            | **45.3%**  | **42.0%**      |
> | **Time (s)** | **18.29**      | 22.63             | 44.39      | 56.22          |
>
> [1, 2, 3] as cited in Official Review by Reviewer Tm5V.
>
> [4] HPR-LP: An implementation of an HPR method for solving linear programming.
>
> [5] The singularity set of optimal transportation maps

---

### Official Review · Reviewer_a2S6 · 2025-10-30

**Soundness:** 3
**Presentation:** 2
**Contribution:** 3
**Rating:** 4
**Confidence:** 3

**Summary:**

The paper proposed a hierarchical algorithm to solve large-scare OT problems. By several techniques including hierarchical expansion, active support update, and GPU based PDHG. The algorithm justifies its efficiency by improving the state-of-the-art baselines and theoretical scale independent iteration complexity bound.

**Strengths:**

By carefully tuning the three approaches in discrete OT techniques, the algorithm justify itself by testing on common data sets.

**Weaknesses:**

1. A well known issue of the hierarchical approach, is the scalability to higher dimensions rather than 2D mesh like topology. The author shall consider a least a remark on extension to higher dimensions.
2. Another issue of the manuscript is the theoretical justification relies on rather strong assumption (4,5), also as the authors mentioned. Since the complexity of HALO is only an upper bound under strong assumptions, the practical superiority of the HALO is not well-explained. The author may need further discuss the structural advantages comparing with other algorithm, for instance Multiscale-OT.

**Questions:**

Beyond the weakness that requires the author to address.
1. The active support in Def. 1 also appears in [1] and even earlier (see references in [1]), will such randomized method improve the HALO?
2. In experiment, the ShortCut is only implemented on CPU, given its low memory cost and rather low runtime in lower resolution, is there theoretical barrier for ShortCut to implement on GPU and yield an even better result?

[1] Xie, Yue, Zhongjian Wang, and Zhiwen Zhang. "Randomized methods for computing optimal transport without regularization and their convergence analysis." Journal of Scientific Computing 100.2 (2024): 37.

---

> ### Author Response · Authors · 2025-11-21
>
> We thank the reviewer for the constructive feedback. We address the specific concerns below, and highlighted the corresponding revisions in **red** and **blue** in the revised manuscript.
>
> > A well known issue of the hierarchical approach, is the scalability to higher dimensions rather than 2D mesh like topology ...
>
> We acknowledge that current hierarchical approaches generally face challenges in high dimensions due to the growth of the neighborhood required for geometric components. We have added a detailed discussion on this in the **General Response (2. Extension to Higher Dimensions)** and included it in the revised **Section4**.
>
> However, we would like to point out that this does not preclude HALO's potential in high-dimensions:
>
> 1. **Proven Performance in 3D**: As detailed in **General Response (1. Extension to Non-Grid Data, Appendix C)** and **Table R2**, we have successfully extended HALO to 3D point clouds. Compared to the SOTA baseline HiRef ($N=2^{18}$), HALO achieves a 2.0$\times$ speedup and ~84% memory reduction, while consistently delivering superior solution quality (gap $\sim 1e-05$ vs. $\sim 5e-01$ on small scales; HiRef has a  **+24.9%** larger gap on large scales).
> 2. **Potential for High Dimensions:** As discussed in the **General Response (2. Extension to Higher Dimensions, Section 4)**, our preliminary research demonstrates that the hierarchical framework can efficiently solve **exact OT problems with** **millions of points in dimensions >1024 on a single GPU**. This suggests a promising path to overcome high-dimensional challenges.
>
> ------
>
> > ... the practical superiority of the HALO is not well-explained. The author may need further discuss the structural advantages comparing with other algorithm, for instance Multiscale-OT.
>
> We explain the structural advantages and practical superiority of HALO compared to Multiscale-OT from the following aspects:
>
> 1. **GPU-based LP Solver:** HALO is the first to effectively integrate a GPU-based first-order LP solver into the hierarchical framework. It breaks the computational bottleneck of solving massive LP subproblems, which is a major limitation for previous CPU-based Multiscale-OT.
>
> 2. To address the precision issues of first-order GPU solvers, we introduce
>
>    two key designs to ensure stability:
>
>    1. **Conservative Active-Set Update:** Instead of aggressively pruning to the support $\mathrm{spt}(x)$, we construct the new active set $\mathcal{N}'$ based on the previous $\mathcal{N}$. This conservative update is the structural key that enables the proof of Theorem 1, providing an interpretable theoretical explanation for the rapid convergence observed in practice.
>    2. **Dual-Violation Correction:** As demonstrated in our ablation study (Table 4 of Paper), dual-violation correction significantly improves performance (>4.0$\times$ speedup) on difficult instances, preventing the algorithm from stagnating during inner iterations.
>
> ------
>
> > The active support in Def. 1 also appears in [1] and even earlier (see references in [1]), will such randomized method improve the HALO?
>
> We thank the reviewer for highlighting this relevant work, which we have added to the **Introduction**: "...Representative methods include **randomized block coordinate descent methods [Xie et al. 2024]** ..."
>
> The randomized active-support selection in [1] represents a viable update strategy. We anticipate that the introduced stochasticity could help the algorithm break potential stagnation during the refinement phase, potentially serving as a complement to our current dual-violation correction.
>
> ------
>
> > In experiment, the ShortCut is only implemented on CPU, ... is there theoretical barrier for ShortCut to implement on GPU and yield an even better result?
>
> There is no theoretical barrier to implementing ShortCut on GPU. We implemented a **ShortCut-GPU (Appendix G)**  using the same framework as HALO. The comparison results are shown below:
>
> **Table R5: Comparison between ShortCut-GPU and HALO (Time in seconds)**
>
> |                  | 512       | 1024      |
> | ---------------- | --------- | --------- |
> | **ShortCut-GPU** | 21.40     | 68.33     |
> | **HALO**         | **11.17** | **27.73** |
>
> **Observations:**
>
> 1. The GPU implementation **indeed brings significant efficiency** gains to ShortCut compared to CPU baselines. This validates our paper's contribution of introducing GPU-based solvers to the hierarchical framework.
> 2. **HALO still outperforms ShortCut-GPU** significantly (~2.5$\times$ faster at $r=1024$). ShortCut's aggressive update is sensitive to the relative lower precision of GPU solvers, leading to stagnation.
> 3. Conclusion: simply porting ShortCut to GPU is insufficient. HALO's **conservative update** (constructing $\mathcal{N}'$ from $\mathcal{N}$) and **dual-violation correction** (Algorithm 3) are essential to stabilize the algorithm and unlock the full potential of GPU-based hierarchical algorithms.

---

> > ### Comment · Reviewer_a2S6 · 2025-11-26
> >
> > Given that a large part of the submission is promised to be updated, I will keep my score for now and wait for the authors to provide a revised manuscript for clearer evaluation.

---

> > > ### Author Response · Authors · 2025-11-27
> > >
> > > We have uploaded a revised manuscript to reflect the changes discussed above. Specifically, the corresponding revisions are highlighted in **red** and **blue**. We respectfully hope that these updates, along with our new experiments and clarifications, have addressed your concerns. We remain available for any further discussion.

---

### Official Review · Reviewer_u6SB · 2025-10-31

**Soundness:** 3
**Presentation:** 3
**Contribution:** 3
**Rating:** 8
**Confidence:** 3

**Summary:**

The paper introduces HALO (Hierarchical Algorithm for Large-scale Optimal Transport), a GPU-based, memory-efficient method for solving large-scale optimal transport (OT) problems in the plane with squared Euclidean cost. HALO combines a multiscale hierarchical framework with a sparse, active-support refinement scheme and a primal–dual hybrid gradient (PDHG) solver to overcome the severe memory and scalability limitations of existing OT solvers. By solving coarser OT problems first and using their solutions to warm-start finer levels, HALO efficiently refines the transport plan while keeping memory usage to  𝑂(r^2), where r is the number of pixels per dimension.

The method also includes a dual-violation augmentation step that improves robustness, and the authors establish a scale-independent iteration bound, proving that each refinement level requires only 𝑂(1) iterations. On the DOTmark benchmark, HALO outperforms state-of-the-art methods such as HOT, ShortCut, and M3S—achieving up to 8.9× speedup and 70% lower GPU memory use for 1024×1024 images, while maintaining comparable or superior accuracy. Overall, HALO demonstrates near-linear runtime scaling and high parallel efficiency, offering a theoretically grounded and practically scalable solution for large-scale OT computation.

**Strengths:**

The paper’s main strengths lie in its combination of theoretical rigor and practical scalability. HALO introduces a hierarchical, GPU-friendly framework that reduces the memory requirement of large-scale optimal transport (OT) from 𝑂(𝑟^4) to 𝑂(𝑟^2), allowing it to handle very high-resolution problems that were previously infeasible. Its coarse-to-fine multiscale design and sparsity-based active support updates enable near-linear runtime scaling, while the factorization-free PDHG solver fully exploits GPU parallelism for efficient computation.

Equally important, the paper provides strong theoretical guarantees and robust empirical validation. The authors prove a scale-independent iteration bound, ensuring that refinement at each level converges in a constant number of steps, and introduce dual-violation augmentation to improve robustness without sacrificing sparsity. Extensive experiments on the DOTmark benchmark confirm that HALO achieves up to 8.9× speedup and 70% memory savings compared to leading solvers, while maintaining high accuracy. Overall, the method is both theoretically elegant and practically impactful, setting a new benchmark for scalable OT computation.

**Weaknesses:**

The main weaknesses of the paper lie in its limited generality and empirical scope. HALO is tailored for 2D optimal transport problems with squared Euclidean cost on regular grids, and both its theoretical guarantees and hierarchical design rely on this structure. As a result, the method’s applicability to higher-dimensional settings, irregular domains, or non-Euclidean costs remains unclear. Moreover, while the algorithm performs impressively on the DOTmark benchmark, its evaluation is confined to this dataset, leaving open questions about robustness and generalization to more diverse or real-world applications.

Another limitation is the dependence on heuristic components such as dual-violation augmentation and Top-K active-set selection, whose performance may vary with parameter choices not deeply analyzed in the paper. The implementation is also technically complex—combining multiscale hierarchy, active-support refinement, and GPU-based PDHG—which could make reproduction or extension challenging for practitioners. Overall, while HALO is methodologically strong and achieves excellent performance, its restricted scope, heuristic tuning, and limited experimental diversity temper its broader applicability.

**Questions:**

The following questions need to be addressed to further improve the quality:

1. transportation cost: if the transportation cost is not the squared Euclidean distance, like L1 distance, will the method be applicable ?
2. dimension : if the problem is not restricted on 2d, but general n dimensional Euclidean space, can the method work ?
3. range: if the support of the target measure is not convex, but a concave Jordan domain, can the active-support refinement  algorithm still work ? In this situation, there will be complicated singularities in the domain, determining the singularity will be challenging.
4. HALO is built on a first -order optimization framework, is it possible to use Newton's method ?

---

> ### Author Response · Authors · 2025-11-21
>
> We thank the reviewer for the positive assessment, particularly for recognizing the “combination of theoretical rigor and practical scalability”. We address the detailed concerns below, and highlight the corresponding revisions in **red** and **green** in revised manuscript.
>
> > The main weaknesses of the paper lie in its limited generality and empirical scope. HALO is tailored for 2D optimal transport problems with squared Euclidean cost on regular grids, and both its theoretical guarantees and hierarchical design rely on this structure. As a result, the method’s applicability to higher-dimensional settings, irregular domains, or non-Euclidean costs remains unclear.
>
> > dimension : if the problem is not restricted on 2d, but general n dimensional Euclidean space, can the method work ?
>
> > transportation cost: if the transportation cost is not the squared Euclidean distance, like L1 distance, will the method be applicable ?
>
> We appreciate these constructive comments regarding the method's scope. We have comprehensively addressed these points in the **General Response 1-3 (highlighted in red in revised paper)**:
>
> - **Extension to Non-Grid Data:** As detailed in **General Response (1. Extension to Non-Grid Data),** we have successfully extended HALO to irregular domains (e.g., point clouds). New experiments on ModelNet10 (2D/3D point clouds) demonstrate that HALO maintains superior efficiency and scalability on non-grid tasks (see **Table R1** and **Table R2**).
> - **Extension to High Dimensions:** Please refer to **General Response (2. Extension to Higher Dimensions)**. Our method can be naturally scaled to dimensions like 3D (**Table R2**). Regarding our ongoing research, extension to higher dimensions is non-trivial but **promising.**
> - **Extension to General Costs:** Please refer to **General Response (3. Extension to General Costs).** We clarify that while HALO is theoretically applicable to general costs, current practical performance is primarily limited by the **sparsity properties of existing GPU solvers** (e.g., for $L_1$ norms) rather than the hierarchical framework itself.
>
> ------
>
> > Moreover, while the algorithm performs impressively on the DOTmark benchmark, its evaluation is confined to this dataset, leaving open questions about robustness and generalization to more diverse or real-world applications.
>
> To demonstrate generalization beyond DOTmark, we evaluated HALO on **FFHQ** and **WikiArt** (detailed in **Appendix I**). **Results (Table R4)** show consistent efficiency, confirming HALO's robustness across diverse datasets.
>
> **Table R4: Results of HALO on Different Datasets**
>
> |           | Resolution | 256  | 512   | 1024  |
> | --------- | ---------- | ---- | ----- | ----- |
> | Times (s) | FFHQ       | 3.24 | 10.79 | 23.68 |
> |           | WikiArt    | 2.74 | 9.13  | 22.14 |
> |           | DOTmark    | 4.31 | 11.17 | 27.73 |
> | Mem (GB)  | FFHQ       | 0.77 | 2.19  | 6.52  |
> |           | WikiArt    | 0.74 | 1.97  | 6.10  |
> |           | DOTmark    | 0.76 | 2.07  | 6.25  |
>
> ---

---

> ### Author Response · Authors · 2025-11-27
>
> > Another limitation is the dependence on heuristic components such as dual-violation augmentation and Top-K active-set selection, whose performance may vary with parameter choices not deeply analyzed in the paper.
>
> We acknowledge the concern regarding the sensitivity of heuristic parameters. We have addressed this comprehensively in the **General Response (4. Sensitivity Analysis of $\beta$, also in Appendix F)**. As detailed in **Table R3**, we evaluated HALO while varying the key hyperparameter $\beta$ across a broad range. The results confirm that dual-violation correction is a key component while both runtime and memory usage remain **highly stable**.
>
> ------
>
> >  The implementation is also technically complex—combining multiscale hierarchy, active-support refinement, and GPU-based PDHG—which could make reproduction or extension challenging for practitioners
>
> To ensure accessibility, **we commit to open-sourcing the full implementation upon acceptance (see Introduction).** The codebase will be well-organized into modular components, and combined with a user-friendly interface, which will greatly facilitate both reproduction and future extensions.
>
> ------
>
> > range: if the support of the target measure is not convex, but a concave Jordan domain, can the active-support refinement algorithm still work ? In this situation, there will be complicated singularities in the domain, determining the singularity will be challenging.
>
> Yes, the algorithm works robustly.
>
> - **Theory Aspect:** Although the shielding component is inspired by Brenier's Theorem[1], the convergence of HALO only relies on discrete optimization. The objective value is **monotonically non-increasing**, guaranteeing convergence regardless of support convexity or singularity sets.
>
> - **Empirical Validation:** HALO successfully handled instances with non-convex supports.
>   - **Non-Grid Data:** The 3D point clouds (ModelNet10, **Table R2**) often sample non-convex manifolds
>   - **DOTmark:** Images in `Shapes` and `Microscopy` often exhibit non-convex structures because of zero-mass pixels. As observed in experiments (Response to Review Tm5V, **Table R7**), while such geometries do slow down convergence, HALO still successfully solves all these instances.
>
> ------
>
> > HALO is built on a first-order optimization framework, is it possible to use Newton's method ?
>
> Yes, it is feasible to use Newton’s method, as demonstrated by works like MSSN[2]. However, the actual performance of MSSN would heavily depend on the implementation efficiency of the CG solver on GPUs, which remains an interesting research direction
>
> [1] Brenier. Polar factorization and monotone rearrangement of vector-valued functions. CPAM 1991.
>
> [2] A Multiscale Semi-Smooth Newton Method for Optimal Transport.

---

### Official Review · Reviewer_rnK4 · 2025-11-02

**Soundness:** 3
**Presentation:** 2
**Contribution:** 3
**Rating:** 6
**Confidence:** 3

**Summary:**

- The authors proposes HALO (Hierarchical Algorithm for Large-scale Optimal Transport), a novel GPU-friendly method for solving large discrete optimal transport (OT) problems efficiently.
- The key idea is to build a coarse-to-fine multi-scale hierarchy and iteratively refine the transport plan while maintaining an active support set that captures potentially non-zero couplings.
- Each refinement step solves a restricted OT problem via a primal-dual hybrid gradient (PDHG) method optimized for GPU computation.
- Experiments on large-scale image OT tasks (DOTmark dataset) demonstrate significant speed and memory savings ove baselines like HOT, ShortCut, and M3S.

**Strengths:**

- The paper successfully solves the scalability bottleneck of OT on high-resolution data with well-designed hierarchical + active-support framework and solid theoretical justification.
- The experimental results are promising with great speed and memory advantage.

**Weaknesses:**

- Currently limited to 2D grid supports with squared Euclidean cost.
- The active-support update relies on heuristic parameters (e.g., β) without sensitivity analysis.
-Experiments only cover image data; no test on non-grid or higher-dimensional problems. Does it work on other distributions?

**Questions:**

Please mainly see the above weakness part

---

> ### Author Response · Authors · 2025-11-21
>
> We thank the reviewer for the positive assessment, acknowledging that our method "successfully solves the scalability bottleneck" and provides "solid theoretical justification". We address the detailed concerns below, and highlight the corresponding revisions in **red** in the revised manuscript.
>
> > Currently limited to 2D grid supports with squared Euclidean cost.
>
> > Experiments only cover image data; no test on non-grid or higher-dimensional problems. Does it work on other distributions?
>
> We appreciate these insightful comments, and we have addressed the extensions comprehensively in **General Response 1, 2, and 3.**
>
> - **Extension to Non-Grid Data (2D/3D, Appendix C):** We demonstrated that grid-specific components can be substituted with $2^d$-trees and KNN. New experiments on ModelNet10 (**Table R1** and T**able R2**) verify its efficiency on irregular domains.
> - **Extension to Higher Dimensions & General Costs (Section 4):** We discussed the promising potential for scaling to much higher dimensions (**General Response, 2. Extension to Higher Dimensions**) and clarified that the current limitation for general costs stems from the sparsity properties of GPU solvers (**General Response, 3. Extension to General Costs**)
>
> ------
>
> > The active-support update relies on heuristic parameters (e.g., β) without sensitivity analysis.
>
> We have addressed this by conducting a sensitivity analysis in **General Response (4. Sensitivity Analysis of $\beta$, Appendix F)**. The results (**Table R3**) confirm that both runtime and memory usage remain highly stable across a broad range of $\beta$, indicating robustness to parameter choices.

---

> > ### Author Response · Authors · 2025-11-27
> >
> > We have uploaded a revised manuscript to reflect the changes discussed above. Specifically, the corresponding revisions are highlighted in **red**. We respectfully hope that these updates, along with our new experiments and clarifications, have addressed your concerns. We remain ready to answer any further questions.

---

### Author Response · Authors · 2025-11-21
**General Response**

We thank the reviewers for their constructive comments, and appreciate their recognition of the significance of our methods. There are several common concerns raised in the reviews, such as extension to non-grid data, higher dimensions, general costs, and the stability of parameters. Therefore in this response, we address the above common issues (revisions highlighted in **red** in the revised manuscript).

### **1. Extension to Non-Grid Data**

We have fully implemented the extension to non-grid data in the revised manuscript.

HALO can be naturally extended to non-grid data, e.g. point clouds, by replacing the grid-specific components with general structures:

1. **Hierarchy (Section 2.2, Appendix C.1)**: Generalize the grid merging to spatial partitioning structures (e.g., 2^d-trees) to build the hierarchical structure .
2. **Shielding (Algorithm 3, Appendix C.2)**: Generalize the 8 neighborhood to K-Nearest Neighbors.

**New Experiments:**

1. **Datasets (Appendix E)**: We constructed a non-grid benchmark using ModelNet10, generating 30 instances of 3D point clouds and their 2D PCA projections.
2. **Baselines (Appendix D):** We compare against **HiRef** (ICML 2025 Oral, SOTA for ultra-large-scale problems [1]) and Sinkhorn (eps=1e-3). For fairness, we employ the default settings provided in official implementations.
3. **Note:** We use uniform marginals since HiRef is limited to the assignment problem. For accuracy, we report the relative objective gap, using standard solver `ot.emd` as ground truth for $n \le 2^{14}$ and HALO as ground truth for $n \ge 2^{15}$.
4. **Results (Section 3.2):** **Table R1 (2D)** and **Table R2 (3D)** report the comprehensive performance from $n=2^{11}$ to $2^{19}$
   1. **Accuracy:** HALO consistently achieves negligible gap compared to `ot.emd` on small scales (**$\sim 10^{-5}$**) and finds significantly lower transport costs than HiRef on large scales, with gaps exceeding **30%** (2D) and **25%** (3D).
   2. **Scalability:** HALO exhibits **near-linear** complexity both in runtime and memory. At $n=2^{18}$, it achieves a **3.47$\times$ (2D) / 1.84$\times$ (3D) speedup** and **~82% (2D) / ~83% (3D) memory reduction** vs. HiRef. Notably, HALO is the **only** solver capable of scaling to $n=2^{19}$.

### **2. Extension to Higher Dimensions**

We acknowledge that our current submission considers only 2D problems. As shown in **Table R1** and **Table R2**, extension to 3D point clouds is also effective. To even higher dimensions, the bottleneck would be shielding-based component in Algorithm 3, which has scalability limitation and becomes inefficient for high-dimension problems.

However, **contrary to common belief** that hierarchical algorithms cannot be extended to high dimensions, our ongoing preliminary results demonstrate that our hierarchical framework can effectively **scale to** **high dimensions (dim>1000) and large scales (n>1M) on a single GPU** by prioritizing the dimension-independent dual-violation correction. While fully implementing this extension involves non-trivial technical challenges beyond the current scope, we have included this promising direction in **Section 4**.

### **3. Extension to General Costs**

We can implement HALO to address general costs, e.g. $L_1$; however, the acceleration would be limited. The main reason is that for $L_1$ costs, the solution returned by GPU solvers (e.g., cuPDLP or HPR-LP [1]) is not as sparse as that from $L_2^2$ cost. As a result, it requires longer time and higher memory, limiting the potential speedup of HALO compared to the current $L_2^2$ setting.

To address this issue, there are two possible choices:

1. develop **GPU-based crossover algorithms** to get sparser solutions;
2. design more flexible active-support strategies.

We had added a remark on it in **Section 4**.

### **4. Sensitivity Analysis of $\beta$**

Regarding the sensitivity of $\beta$, we emphasize that HALO is very robust to the choice of $\beta$. To demonstrate, we test HALO on the DOTmark dataset with $\beta=2^{-4,-3,-2,-1,0}$ and $\beta=0$, see **Table R3** for result:

1. For $\beta=2^{-3,-2,-1,0}$, the runtimes are quite close. For $r=1024$, the time difference is less than 17%. Memory usage is virtually unaffected by $\beta$, staying constant at $\sim 6.3$ GB.
2. For $\beta$=0, which corresponds to removing the dual-violation augmentation, we observe a sharp performance drop.

These results confirm that the dual-violation component is crucial for efficiency, but the algorithm is insensitive to the choice of $\beta$ within a wide range. We recommend $\beta = 2^{-2}$ as a safe default, and we have  added this analysis in **Appendix F**.

[1] Halmos et al. Hierarchical Refinement: Optimal Transport to Infinity and Beyond.

[2] HPR-LP: An implementation of an HPR method for solving linear programming. Mathematical Programming Computation 2025.

---

> ### Author Response · Authors · 2025-11-21
>
> **Table R1: Performance on Non-Grid 2D Data (ModelNet10-PCA).**
>
> | Metric      | Method          | 2^11          | 2^12          | 2^13          | 2^14          | 2^15      | 2^16      | 2^17      | 2^18      | 2^19      |
> | ----------- | --------------- | ------------- | ------------- | ------------- | ------------- | --------- | --------- | --------- | --------- | --------- |
> | **Time(s)** | **HALO (ours)** | **1.79**      | **2.89**      | **4.73**      | **7.82**      | **12.50** | **23.44** | **53.25** | **121.6** | **247.0** |
> |             | HiRef           | 3.36          | 6.70          | 11.89         | 23.4          | 46.4      | 94.1      | 190.4     | 422.0     | OOM       |
> |             | Sinkhorn        | 2.81          | 3.54          | 8.70          | 28.5          | OOM       | OOM       | OOM       | OOM       | OOM       |
> | **Mem(GB)** | **HALO (ours)** | **0.50**      | **0.54**      | **0.52**      | **0.55**      | **0.62**  | **0.74**  | **1.08**  | **1.92**  | **3.97**  |
> |             | HiRef           | 0.84          | 0.85          | 0.86          | 0.90          | 1.02      | 1.65      | 3.60      | 10.89     | OOM       |
> |             | Sinkhorn        | 0.52          | 1.00          | 2.92          | 10.60         | OOM       | OOM       | OOM       | OOM       | OOM       |
> | **Gap**     | **HALO (ours)** | **+1.83e-06** | **+3.83e-05** | **+4.70e-05** | **+5.68e-05** | -         | -         | -         | -         | -         |
> |             | HiRef           | +6.77e-01     | +6.49e-01     | +5.48e-01     | +4.37e-01     | +3.82e-01 | +4.04e-01 | +3.00e-01 | +3.25e-01 | OOM       |
> |             | Sinkhorn        | +3.67e-02     | +4.44e-02     | +4.69e-02     | +4.02e-02     | OOM       | OOM       | OOM       | OOM       | OOM       |
>
>
>
> **Table R2: Performance on Non-Grid 3D Data (ModelNet10).**
>
> | Metric      | Method          | 2^11          | 2^12          | 2^13          | 2^14         | 2^15      | 2^16      | 2^17      | 2^18      | 2^19      |
> | ----------- | --------------- | ------------- | ------------- | ------------- | ------------ | --------- | --------- | --------- | --------- | --------- |
> | **Time(s)** | **HALO (ours)** | **1.64**      | 4.63          | **8.38**      | **15.54**    | **26.25** | **47.42** | **88.51** | **229.7** | **444.3** |
> |             | HiRef           | 3.39          | 6.41          | 11.88         | 23.6         | 46.4      | 94.4      | 189.9     | 422.7     | OOM       |
> |             | Sinkhorn        | 2.89          | **3.66**      | 8.832         | 29.5         | OOM       | OOM       | OOM       | OOM       | OOM       |
> | **Mem(GB)** | **HALO (ours)** | **0.48**      | **0.50**      | **0.54**      | **0.54**     | **0.64**  | **0.77**  | **1.12**  | **1.83**  | **2.99**  |
> |             | HiRef           | 0.85          | 0.85          | 0.86          | 0.91         | 1.03      | 1.67      | 3.60      | 10.92     | OOM       |
> |             | Sinkhorn        | 0.52          | 1.00          | 2.92          | 10.60        | OOM       | OOM       | OOM       | OOM       | OOM       |
> | **Gap**     | **HALO (ours)** | **+1.32e-06** | **+1.76e-05** | **+4.22e-05** | **5.33e-05** | -         | -         | -         | -         | -         |
> |             | HiRef           | +5.55e-01     | +5.18e-01     | +4.78e-01     | +4.19e-01    | +3.51e-01 | +3.13e-01 | +2.77e-01 | +2.49e-01 | OOM       |
> |             | Sinkhorn        | +1.75e-02     | +2.26e-02     | +2.70e-02     | +3.20e-02    | OOM       | OOM       | OOM       | OOM       | OOM       |
>
>
> **Table R3: Sensitivity Analysis of $\beta$ on DOTmark.** Performance remains stable for $\beta \in [2^{-3}, 2^0]$. Setting $\beta=0$ (disabling correction) leads to ~2x slowdown, justifying the necessity of the module.
>
> | Metric       | $\beta$       | 64       | 128      | 256      | 512      | 1024      |
> | ------------ | ------------- | -------- | -------- | -------- | -------- | --------- |
> | **Time (s)** | $2^0$         | 0.77     | 1.18     | 2.79     | 9.24     | 28.86     |
> |              | $2^{-1}$      | 0.78     | 1.12     | 2.70     | 8.76     | 25.19     |
> |              | $2^{-2}$      | 0.78     | **1.11** | **2.62** | 8.84     | 25.42     |
> |              | $2^{-3}$      | **0.72** | **1.11** | 2.71     | **8.72** | **24.67** |
> |              | $2^{-4}$      | 1.73     | 2.42     | 4.44     | 11.90    | 29.64     |
> |              | $0$ (No dual) | 1.90     | 2.99     | 5.47     | 15.23    | 52.85     |
> | **Mem (GB)** | $0\sim 2^{0}$ | ~0.39    | ~0.48    | ~0.76    | ~2.10    | ~6.30     |

---

### Author Response · Authors · 2025-12-02
**Summary of Revisions for AC**

# Appreciation and Core Contribution

We sincerely appreciate the Area Chairs' time and dedication to the community. We also thank the reviewers (rnK4, u6SB, a256, Tm5V) for their constructive feedback, highlighting our work's 'theoretical rigor' and ability to 'successfully solve the scalability bottleneck'.

**Key Insight**: HALO demonstrates a promising direction to **achieve state-of-the-art scalability in both computational speed and memory efficiency for Optimal Transport, entirely without the need for regularization or approximation.**
This is realized by synergizing a hierarchical framework, sparse active-support refinement, and a GPU-friendly LP solver. Consequently, HALO delivers up to 8.9x speedup and 70.5% memory reduction on 2D images , backed by a proven scale-independent iteration complexity.

# Common Concern

We have comprehensively addressed the common concerns raised by reviewers regarding extensions and stability:

## **1. Extension to Non-Grid Data**

As requested, we successfully extended HALO to **non-grid data** (e.g., point clouds).

**Evidence:** New experiments on ModelNet10 (**Tables R1, R2**) demonstrate that HALO outperforms the SOTA solver. Notably, at $N=2^{18}$ (3D), HALO achieves a 1.84x speedup and 83.2% memory reduction, while significantly finding a 24.9% lower transport cost.

## **2. Extension to High Dimensions**

**While hierarchical algorithms are commonly believed to face bottlenecks in high dimensions, our framework offers a counter-intuitive perspective:**

- Proven Feasibility: Extension to moderate dimensions like 3D is fully feasible, as evidenced by our new experiments on ModelNet10 (**Table R2**)
- **Path to High Dimensions:** While the shielding-based component drives our superior performance in moderate dimensions, the **dual-violation component is notably dimension-independent**. Our ongoing research suggests that by further optimizing the latter, the framework shows potential to scale to high dimensions ($>1000$), promising to greatly alleviate the scalability bottleneck for large-scale exact OT. We have clarified the current scope in the Introduction and added this promising direction to Section 4.

## **3. Extension to General Costs**

We clarify that HALO is theoretically applicable to general costs (e.g., $L_1$), but the current practical limitation lies in the sparsity of solutions returned by existing GPU LP solvers for non-$L_2^2$ costs. We have outlined clear paths (e.g., GPU-based crossover algorithms, more flexible active-support strategies) to address this in Section 4.

## **4. Parameter Sensitivity**

We conducted a sensitivity analysis on $\beta$ (**Table R3**) as requested.

**Result:** Performance is highly stable (fluctuation $<20\%$) across $\beta \in [2^{-3}, 2^0]$. Conversely, disabling the component ($\beta=0$) causes significant slowdowns (~2x), validating the necessity of our dual-violation correction.

# Specific Concerns

Beyond common concerns, we addressed individual reviewers' questions with expanded evaluations.

- **Generalization across Domains (Reviewer u6SB, Table R4):** We validated HALO on diverse datasets beyond DOTmark, including FFHQ and WikiArt, confirming consistent efficiency.
- **Structural Superiority over Multiscale OT (Reviewer a256, Table R5):** We implemented a GPU version of ShortCut as requested. HALO still outperforms ShortCut-GPU by ~2.5x, proving that our conservative active-set strategy and dual-violation correction, which constitute distinct structural advantages over previous Multiscale-OT methods, are the key drivers of performance, rather than merely GPU implementation.
- **Solver Flexibility (Reviewer Tm5V, Table R6) :** To address concerns about reliance on cuPDLP, we integrated HALO with HPR-LP. Results confirm HALO maintains its advantage (6.1x speedup over M3S) regardless of the underlying solver.
- **Handling and Understanding Geometric Complexity (Reviewer u6SB and Tm5V, Table R7):** We analyzed performance across image classes of DOTmark. While pixel intensity sparsity induces singularities that naturally increase runtime (consistent with our theory), HALO robustly solves all instances.
- **Reproducibility (Reviewer u6SB)**: We committed to open-sourcing our full implementation to facilitate future research.

---

### Meta-Review · Area_Chair_ZjaC · 2025-12-28

**Summary:**

This paper proposes a hierarchical algorithm for large-scale OT (HALO) by synergizing GPU-based linear program (LP) solver with a multi-scale representation of the OT problem. HALO is memory efficient and has the potential to scale to large 2D and 3D OT problems. Empirical studies demonstrate HALO achieves sizable speedup and significant memory savings compared to leading exact OT solvers.

The reviewers expressed several merits of the proposed HALO algorithm including its scalability to high-resolution data, solid theoretical support, and convincing experimental results in speed and memory.

The reviewers also raised several common weakness and questions:

- limitation to OT problems with squared Euclidean cost on regular grid;
- possibility of 3D OT problems;
- HALO dependence on heuristic components such as dual violation step;
- lack of sensitivity analysis.

**Reviewer Concerns:**

In response, the authors conducted more numerical experiments to address those concerns and clarified some points. In particular, the paper was revised to include extension of HALO to non-grid data with new experiments and a sensitivity check on the hyperparameters in dual-violation correction.

**Reviewer Scores:**

4 reviewers submitted their comments and scores (6/4/6/8) with confidence (3/3/3/3), with average score 6 and average confidence 3.

Given the amount of the extra numeric evidence in authors’ rebuttal, I expect that if the discussions were continuing, the reviewers would have maintained an overall positive consensus of this paper.

---

### Decision · Program_Chairs · 2026-01-26

Accept (Poster)